# A mechanistic account of serotonin's impact on mood

Jochen Michely[1,2,3 ✉], Eran Eldar[1,4], Ingrid M. Martin[2] & Raymond J. Dolan [1,2]

Selective serotonin reuptake inhibitors (SSRIs) constitute a first-line antidepressant intervention, though the precise cognitive and computational mechanisms that explain treatment response remain elusive. Using week-long SSRI treatment in healthy volunteer participants, we show serotonin enhances the impact of experimentally induced positive affect on learning of novel, and reconsolidation of previously learned, reward associations. Computational modelling indicated these effects are best accounted for by a boost in subjective reward perception during learning, following a positive, but not negative, mood induction. Thus, instead of influencing affect or reward sensitivity directly, SSRIs might amplify an interaction between the two, giving rise to a delayed mood response. We suggest this modulation of affect-learning dynamics may explain the evolution of a gradual mood improvement seen with these agents and provides a novel candidate mechanism for the unfolding of serotonin's antidepressant effects over time.

[1] Max Planck UCL Centre for Computational Psychiatry and Ageing Research, University College London, London, UK. [2] Wellcome Centre for Human Neuroimaging, University College London, London, UK. [3] Department of Psychiatry and Psychotherapy, Charité - Universitätsmedizin Berlin, Berlin, Germany. [4] Psychology and Cognitive Sciences Departments, Hebrew University of Jerusalem, Jerusalem, Israel. ✉email: jochen.michely@charite.de

Major depression is a leading cause of disability world-wide, representing one of society's most challenging, and costly, health burdens[1]. Pharmacotherapy of depression primarily targets monoamine neurotransmitter function, with selective serotonin reuptake inhibitors (SSRIs) constituting first-line antidepressant intervention[2]. Despite their well-described clinical[3] and neurochemical effects[4,5], the cognitive and computational mechanisms underlying SSRI treatment effectiveness remain elusive[6,7]. This lack of a mechanistic understanding arguably compromises development of effective treatment stratification and clinical prediction models[8].

At the cognitive level, a bias towards attending and remembering negative information[9], as well as a decrease in perception of positive information[10,11], are considered to play a key role in both the development and maintenance of depression. Consequently, modulating affective biases, with the aim of boosting reward exposure and positive experience, is a principled target in psychological interventions, as in the paradigmatic instance of behavioural activation therapy[12]. Intriguingly, a growing body of work suggests pharmacological intervention might influence affective processing in a similar manner, in other words via a boost in perception of positive affective information that induces a 'rosier' perspective[7,13]. Nevertheless, it remains unclear how a pharmacological induction of affective biases in perceptual processing, for example as reported in emotional face perception, translates into overt mood change[14–16]. Influential theories propose affective biases may influence cognitive processes other than perception, such as reward learning and affective memory consolidation[17,18].

We previously proposed a mechanistic framework for how affect-learning interactions contribute to mood dynamics[19]. Empirically, we have shown how unexpected outcomes alter affective state, which in turn biases a sensitivity to subsequent reward outcomes[20,21]. Theoretically, this two-way relationship can set in train an escalating positive feedback loop wherein good outcomes improve mood which, in turn, improves perception of subsequent outcomes leading to further mood elevation.

Based upon clinical observation that antidepressants do not enhance mood either acutely or directly, but are instead characterised by a delayed clinical onset[22], we hypothesised SSRIs would influence an integral aspect of this feedback loop. Specifically, we conjectured serotonin would boost the impact of positive affect on reward processing, an effect that can give rise to gradual mood improvement. To test our hypothesis, we used a week-long SSRI treatment intervention in healthy volunteers, with repeated experimental testing, to probe two specific predictions. First, we predicted that, during learning of novel reward associations, SSRIs would boost the impact of positive affect on the subjective value of reward. Second, we predicted that, at the time of exposure to previously learned reward associations, SSRIs would boost the impact of positive affect on value reconsolidation.

## Results

**SSRIs, positive mood and learning.** Sixty-four healthy volunteers were randomly allocated into two equal-sized groups (Supplementary Table 1) and asked to take a daily oral dose of either the SSRI citalopram (20 mg) or placebo over 7 consecutive days. Subjects performed four laboratory testing sessions, comprising a pre-drug session on day 1 as well as three on-drug sessions on days 1, 4, and 7, respectively (Fig. 1d).

On every session, subjects performed two blocks of a trial-and-error reward learning task, during which they were asked repeatedly, via an experiential sampling protocol, to report their current emotional state by responding to the question 'How do you feel right now?' (Fig. 1c). On every trial (Fig. 1a), subjects chose between two images presented on screen and received a monetary reward ('gold coin', £0.03), or no reward ('red cross'),

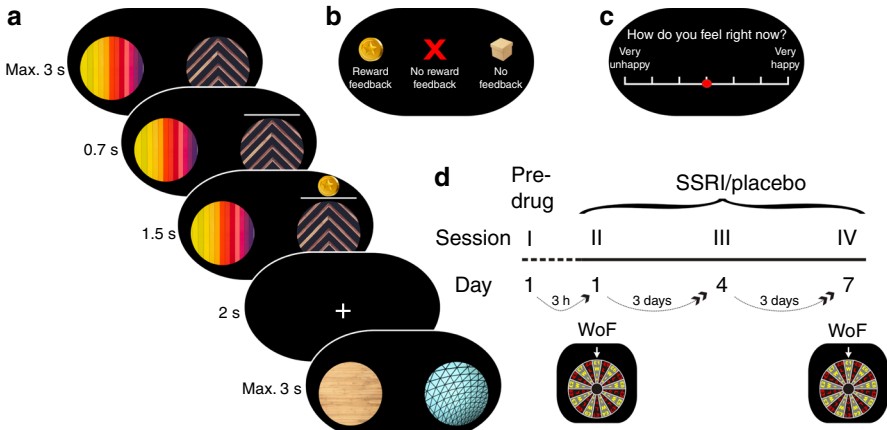

**Fig. 1 Experimental task and study design. a** Trial structure. On every trial, subjects had 3 s to make a choice between two presented images. The chosen image was highlighted for 0.7 s. Subsequently, an outcome, either reward (gold coin), no reward (red cross), or no feedback (box), was shown above the image for 1.5 s. After a 2 s fixation cross period, the next trial started. **b** Different outcome types. On learning trials, subjects encountered either a monetary reward (gold coin, £0.03), or no reward (red cross) outcome, depending on the chosen image's associated reward probability. Here, each image was associated with either a fixed high (70%) or a fixed low reward probability (30%). On test and interleaved maintenance trials, outcomes were not revealed. Instead, they were 'hidden in a box' visible to subjects. Subjects were instructed that images still led to reward outcomes, depending on reward probabilities. However, to prevent new learning at this stage, feedback was not shown. **c** Mood rating. Multiple times per block (three times during blocks of 48 trials, four times during blocks of 72 trials; fixed intervals), subjects were asked to report their current emotional state by responding to the question 'How do you feel right now?'. The rating was provided without time restriction by moving a red cursor that always started at the midpoint along a scale ranging from 'very unhappy' to 'very happy'. **d** Study design. Subjects were randomly allocated to take a daily dose of 20 mg citalopram or placebo for 7 days and participated in four session over the week: a pre-drug session I on day 1, and three sessions on the drug, on days 1, 4, and 7 (sessions II, III and IV). To modulate subject's affective state between learning blocks, a Wheel of Fortune (WoF) draw was held in session II (day 1) and IV (day 7). Cf. Supplementary Fig. 1 for details on session structure and image appearance across the entire study.

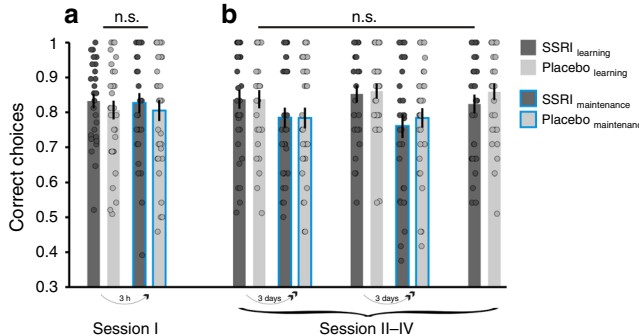

**Fig. 2 No drug effects on reward learning and maintenance of learned associations.** Correct choices for images learned during **a** the pre-drug session and **b** over the course of 1-week drug treatment. Task performance was quantified as proportion of choices of the image associated with a high reward probability on trials comparing images of different reward probability, i.e., high vs. low. Learning was assessed during a test block on every session, comparing images from blocks 1 and 2, encountered during preceding learning. Maintenance was assessed during interleaved trials involving familiar images on subsequent sessions, where images learned about in session I re-appeared in session II, and images from session II in session III, etc. Notably, both learning and maintenance were assessed on choices between the same image pairs, with the time-lag between learning and maintenance test constituting the only difference. Results show that learning performance was stable across sessions, and unaffected by SSRI treatment (pre-drug ($n = 63$): $F_{1,61} = 0.5$, $p = 0.457$; sessions II–IV ($n = 62$): drug: $F_{1,60} = 0.6$, $p = 0.428$; drug × session: $F_{2,120} = 0.3$, $p = 0.717$). Moreover, performance decreased more for 3-day (II to III and III to IV) as compared to 3-h time-lag (I to II, time: $F_{2,120} = 6.8$, $p = 0.002$), but performance decline was unaffected by drug manipulation drug: ($F_{1,60} = 0.03$, $p = 0.858$; drug × time: $F_{2,120} = 0.005$, $p = 0.995$). n.s. = not significant (no difference across drug groups). Error bars indicate SEM.

depending on the chosen image's associated reward probability. Each learning block comprised a total of four novel images, two with fixed high (70%) and two with fixed low reward probability (30%). To perform well, and maximise monetary outcome, subjects needed to choose images they considered more likely to lead to reward.

Upon completion of the two learning blocks, subjects performed a final test block involving presentation of all images encountered previously during learning. This design feature allowed us to measure subjects' learning performance, quantified as the proportion of chosen high probability compared to low reward probability images, i.e., high vs. low. In this test block, outcomes were not revealed to prevent further new learning at this stage. Instead, outcomes were 'hidden in a box' visible to subjects (Fig. 1b). We found participants learned to dissociate high and low reward probability images in the pre-drug session, with no between-group difference ($F_{1,61} = 0.5$, $p = 0.457$; Fig. 2a). Moreover, performance did not differ between SSRI and placebo during the drug period (drug: $F_{1,60} = 0.6$, $p = 0.428$; drug × session: $F_{2,120} = 0.3$, $p = 0.717$, Fig. 2b). This indicates that probabilistic reward learning was stable across sessions and unaffected by SSRI treatment.

To generate large prediction errors likely to be sufficient to impact subjects' affective state between learning blocks, we held a Wheel of Fortune (WoF) draw in sessions II (day 1) and IV (day 7; Fig. 3a). The WoF, depicting possible win or loss outcomes in the range of £1−£9, turned gradually for approx. 45 s before stopping. An indicator then revealed an outcome. Each subject won and lost once, with the order counterbalanced across subjects, an amount that randomly varied from £7 to £9, which was added to or subtracted from their overall payment.

A comparison of pre- and post-WoF mood (mean happiness ratings across entire block) indicated that WoF outcome profoundly impacted emotional state, i.e., subjects reported feeling happier in the post-WoF block after a win, and less happy after a loss ($F_{1,62} = 42.0$, $p < 0.001$). Here, the absolute (negative) effect of a loss was significantly greater than the

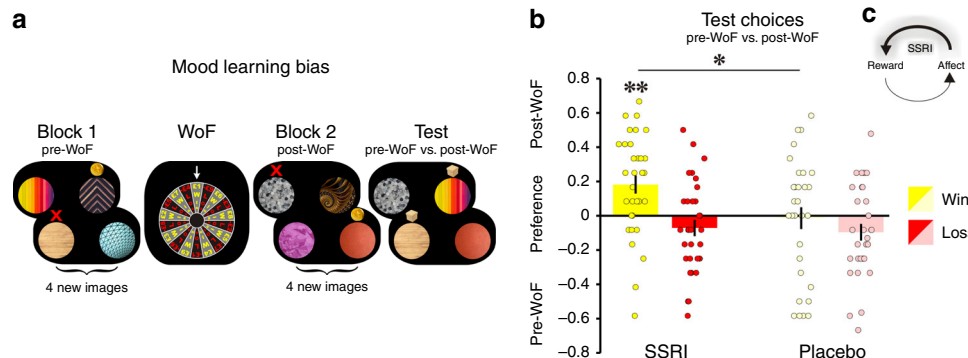

**Fig. 3 SSRIs induce an impact of positive mood on learning. a** Each session comprised two learning blocks, where subjects encountered four novel images, two with a high (70%) and two with a low reward probability (30%). Feedback enabled subjects to learn about images' reward probabilities through trial-and-error. Additionally, sessions II and IV comprised a WoF draw between learning blocks, where each subject either won (yellow) or lost (red) a large monetary amount (£7−£9). After the second learning block, subjects performed a test choice block, where they chose between images from the two preceding learning blocks. To prevent further learning, outcomes were not revealed in these test blocks, and subjects were instructed to rely solely on what they had learned previously about the images. **b** Proportion of test block choices between images encountered during preceding learning with feedback in different blocks, i.e., pre-WoF or post-WoF. This analysis compared choices between images with equal reward probabilities, i.e., high vs. high and low vs. low, and preference is computed as difference between percent choices of post-WoF images and pre-WoF images, as a function of WoF outcome. Participants showed a preference for images encountered in better mood, favouring post-WoF images after a winning draw, and favouring pre-WoF images after a losing draw. The SSRI group showed a significantly positive mood bias ($F_{1,32} = 12.1$, $p = 0.001$), and it was enhanced as compared to placebo subjects ($F_{1,60} = 5.5$, $p = 0.021$, $n = 62$). ** $p < 0.01$, * $p < 0.05$. Error bars indicate SEM. **c** Interaction between affect and reward sensitivity. Rewards impact on emotional state, which, in turn, impacts on perception of subsequent rewards. In this bidirectional interaction, SSRIs enhance the impact of positive affect on reward sensitivity. This serotonergic effect can engender a positive feedback loop.

absolute (positive) effect of a win ($F_{1,62} = 23.5$, $p < 0.001$), a result that has echoes with the notion of loss aversion, where a loss is known to be more impactful than an equivalent gain[23]. However, these mood responses were unaffected by SSRI treatment (drug: $F_{1,62} = 0.8$, $p = 0.361$), meaning that neither the mood change following a WoF win ($F_{1,62} = 0.5$, $p = 0.482$) nor the mood change following a WoF loss ($F_{1,62} = 0.4$, $p = 0.499$) differed between groups. Thus, SSRIs did not affect the manifestation of loss aversion on subjective mood ratings.

Next, we examined whether the WoF impacted on reward sensitivity while subjects were experiencing this altered emotional state. To quantify this, we assessed a preference for images initially encountered before or after the experimental mood manipulation. This preference was computed during the final test block, as the proportion of chosen images encountered previously across different learning blocks, pre-WoF or post-WoF (Fig. 3a). Note this analysis compared image pairs of equal reward probability, i.e., high vs. high and low vs. low.

Participants showed a preference for images encountered previously when in a better mood, favouring post-WoF images after a winning draw, and favouring pre-WoF images after a losing draw ($F_{1,60} = 9.1$, $p = 0.004$; Supplementary Fig. 2A). This suggests elevated mood after winning on the WoF boosted a preference for images encountered in a high mood, whereas loss induced mood change led to a disinclination toward images encountered in a low mood.

Critically, this mood learning bias was modulated by the drug. Subjects on SSRIs showed an enhanced positive mood bias compared to placebo subjects ($F_{1,60} = 5.5$, $p = 0.021$), whereas the negative mood bias was not different between groups ($F_{1,60} = 0.1$, $p = 0.719$). A significant interaction between valence of mood manipulation and drug ($F_{1,60} = 4.4$, $p = 0.038$; Fig. 3b) indicates that SSRI treatment specifically boosted a preference for images encountered following a winning draw, i.e., images learned in a positive mood.

To rule out a possibility that results were driven by primacy (preference for block 1) or recency (preference for block 2) effects, we assessed test block choices from the mood-neutral session III (preference for blocks 1 or 2, in the absence of a WoF draw). We found no difference between treatment groups ($F_{1,60} = 0.5$, $p = 0.470$). Secondly, mood-dependent drug effects from sessions II and IV remained significant when regressing out effects from the mood-neutral session III (win: $F_{1,59} = 6.9$, $p = 0.011$; loss: $F_{1,59} = 0.6$, $p = 0.807$; drug × valence: $F_{1,59} = 4.8$, $p = 0.032$), indicating that the reported findings are independent of primacy or recency effects.

In addition to comparing directly between images with equal reward probabilities, we conducted a more complete and individualised analysis of subjects' test choices, including choices between images with both equal and different reward probabilities. For this, we used a logistic regression, on each session that comprised a WoF draw, that accounted not only for the impact of a mood manipulation, but also for the number of rewards subjects received for choosing each image with feedback during learning. Specifically, choice on test block trials (without feedback) was the dependent variable in the regression model on each session. Difference in number of rewards received (during preceding learning with feedback) for the two images shown on the screen, and a WoF outcome that preceded learning about each image, served as predictor variables. As expected, subjects were more likely to choose images for which they received more rewards during preceding learning ($\beta_1$: $F_{1,60} = 171.2$, $p < 0.001$), but this reward impact on choice was not affected by drug ($F_{1,60} = 0.1$, $p = 0.659$; Supplementary Fig. 3A). In line with the average results presented above, we found in addition a significant effect of mood bias across the entire sample ($\beta_2$: $F_{1,60} = 4.0$, $p = 0.049$),

reflecting subjects' preference for images they had encountered in a positive mood state after a win, and disinclination for images they had encountered in a negative mood state after a loss, an effect not driven by differences in experienced reward. Critically, this effect was modulated by drug group (drug × valence: $F_{1,60} = 4.1$, $p = 0.047$), an effect driven by a significantly enhanced mood bias on the session including a WoF win ($\beta_{1positive}$: $F_{1,60} = 5.5$, $p = 0.022$), but not on the session including a WoF loss ($\beta_{2negative}$: $F_{1,60} = 0.1$, $p = 0.789$) under SSRI as compared to placebo (Supplementary Fig. 3B).

To rule out a possibility that our results were affected by how often subjects had sampled images during learning, we conducted two further control analyses. Firstly, we ruled out any difference between treatment groups in their sampling of images with high or low reward probability across sessions ($F_{1,62} = 0.00$; $p = 0.984$). Secondly, we computed a logistic regression analysis to assess whether the number of times an image had been chosen during learning affected how often subjects chose it subsequently during the test block. Here, we found a found a small, yet significant effect ($F_{1,62} = 8.7$; $p = 0.005$), indicating subjects preferred images which they had sampled more frequently during learning and therefore had higher certainty about (over and above the number of experienced rewards per image). Crucially, this measure was unaffected by drug treatment ($F_{1,62} = 0.2$; $p = 0.692$), indicating that SSRI treatment did not impact uncertainty aversion.

**Computational modelling: SSRIs boost reward sensitivity in positive mood.** Next, we used computational modelling to probe the precise learning mechanisms by which SSRIs boost a positive mood bias. A wide range of evidence indicates human and nonhuman animal learning is driven by reward prediction error signals, reflecting a difference between expectations and actual outcomes[24]. We compared a range of reinforcement learning models (cf. Methods for details) in terms of how well each model fitted subjects' choices. The best-fitting model ('dynamic mood model'; Supplementary Fig. 4) accounts for the effect of mood on learning by means of two parameters, one for positive ($M_{positive}$), and one for negative mood ($M_{negative}$), that bias the subjective value of reward following the WoF mood manipulation. Rather than learning from actual reward, this model computes prediction errors with respect to subjective reward, which is up- or down-regulated depending on mood state.

When individually fitting the model parameters to each subject's choices, we found that model parameter estimates strongly correlated with model-free results (Supplementary Fig. 5). Additionally, generating simulated data based upon these parameters showed that the model captured core features of the real data (Supplementary Fig. 6), that model comparison could identify the model that generated the data (Supplementary Fig. 7), and that individual parameter estimates could be accurately recovered (Supplementary Fig. 8).

Individually fitting the model parameters to each subject's choices showed no difference between drug and placebo group for learning and choice parameters (dynamic learning rate ($\varepsilon$): $F_{1,62} = 0.2$, $p = 0.589$; decision temperature ($\beta$): $F_{1,62} = 1.2$, $p = 0.276$). However, we found a significant difference in mood bias (drug × valence: $F_{1,60} = 5.1$, $p = 0.026$), driven by an elevated positive ($F_{1,60} = 4.3$, $p = 0.041$) but not negative mood bias parameter ($F_{1,60} = 0.6$, $p = 0.435$) in SSRI as compared to placebo-treated subjects (Supplementary Fig. 9). This is consistent with the idea that rewards are subjectively perceived as greater in a positive mood, an effect that was significantly enhanced in the SSRI treatment group.

**SSRIs modulate the impact of mood on value reconsolidation**. In addition to novel images, each learning block comprised interleaved trials where subjects made choices between images that were familiar from the previous session. Similar to the test block, outcomes were not revealed on such trials so as to prevent further learning, entailing subjects needed to rely on previously acquired knowledge when making choices. This interleaving of trials allowed us to test how accurately subjects maintained learned reward associations over time, by quantifying the proportion of high probability image choices compared to low reward probability images, high vs. low. We found a significant effect of time-lag between initial learning and our test for maintenance. Specifically, performance decreased more for 3-day (II to III and III to IV) as compared to 3-h time-lag (I to II; time: $F_{2,120} = 6.8$, $p = 0.002$; Fig. 2). However, performance decline was unaffected by drug manipulation (drug: $F_{1,60} = 0.03$, $p = 0.858$; drug × time: $F_{2,120} = 0.005$, $p = 0.995$; Fig. 2), suggesting SSRI treatment did not impact on short-term maintenance of learned reward associations.

An interleaving of trials without feedback allowed us to assess a second prediction related to an impact of SSRIs on how mood affects reconsolidation of previously acquired reward associations. For this purpose, a subset of images re-appeared on no-feedback trials that had been presented either before, or after, the WoF draw. In the final test block, subjects were then asked to choose between pre-WoF vs. post-WoF images of equal reward probability (Fig. 4a). By design, such images were identically affected by time-lag and initially learned in the same block, without any biasing influence of mood (during the previous session). The images differed only in relation to the mood state under which previously acquired reward associations were subject to reconsolidation during choice (on the current session).

Across the entire sample, subjects preferred images that re-appeared post-WoF, irrespective of whether they were linked to a win or loss (assessed on equal reward probability trials; $F_{1,60} = 7.5$, $p = 0.008$; Supplementary Fig. 2b). This raises a possibility that increased arousal, engendered by the outcomes of the WoF draw, enhanced the subjective value of images during post-WoF

reconsolidation, inducing a consequential preference for these images. This points to the possibility of a qualitatively distinct effect of the WoF draw on learning and on reconsolidation (Supplementary Fig. 2 for effects across the entire sample), a difference that is further supported by an absence of any relationship between the two effects (win: $R = 0.07$, $p = 0.705$; loss: $R = -0.18$, $p = 0.303$).

Critically, an arousal bias was differentially modulated by treatment (drug × valence: $F_{1,60} = 4.8$, $p = 0.032$; Fig. 4b), an effect in the SSRI group driven by a significant preference for images encountered after winning, but not after losing, (win: $F_{1,15} = 6.4$, $p = 0.023$; loss: $F_{1,15} = 1.4$, $p = 0.246$); and in the placebo group after losing but not winning (win: $F_{1,15} = 0.5$, $p = 0.819$; loss: $F_{1,15} = 10.1$, $p = 0.006$).

We found no evidence that subjects performed more accurately (assessed on high vs. low probability trials), as a function of the draw (post-WoF vs. pre-WoF: $F_{1,62} = 1.1$, $p = 0.302$), as a function of valence of the draw (post-WoF win vs. post-WoF loss: $F_{1,60} = 0.4$, $p = 0.538$), or as a function of drug condition (post-WoF drug: $F_{1,60} = 0.03$, $p = 0.864$, post-WoF drug × valence: $F_{1,60} = 0.7$, $p = 0.417$). These observations rule out an interpretation based upon images being remembered more accurately either as a function of mood induction or of drug condition.

Thus, the findings are consistent with the WoF draw amplifying the subjective value of images re-encountered in the post-WoF block, leading to a biased choice preference towards such images. As predicted, SSRI treatment induced a preference for images re-encountered in a positive, compared to a negative, mood state.

**SSRIs induce delayed mood effects consistent with affect-learning interaction**. Next, we tested two simpler alternative hypotheses concerning how SSRIs could elevate mood, which do not require affect-learning interactions. First, SSRIs could simply elevate positive affect directly and, if so, we would expect this to be reflected in changes in mood ratings measured during the first block of our experimental task, i.e., prior to an actual mood

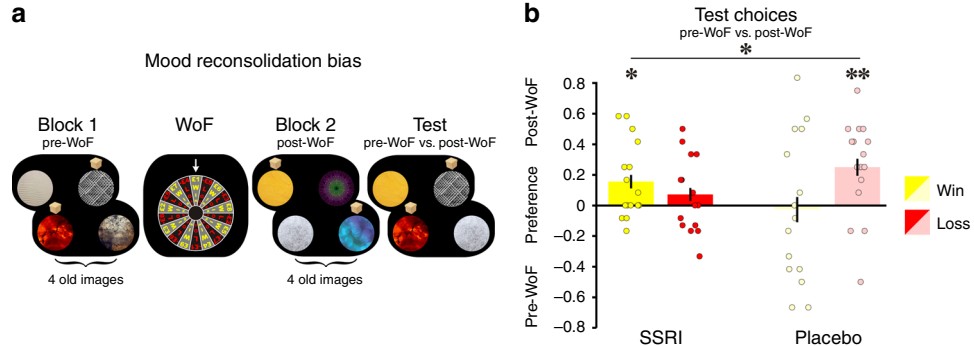

**Fig. 4 SSRIs modulate the impact of mood on reconsolidation. a** In addition to novel images, each block comprised interleaved trials involving choices between familiar images from previous sessions. Outcomes were not revealed on such trials to prevent further learning, and subjects had to rely on previously acquired knowledge when making choices. On session IV, a subset of images re-appeared as interleaved trials either before (pre-WoF), or after (post-WoF), the mood manipulation. The final test block comprised an additional set of trials comparing these images (re-appeared pre-WoF vs. re-appeared post-WoF). By design, such images only differed in the mood state under which subjects retrieved learned reward associations from memory when choosing. Note these images had initially been learned in the same block, unbiased by mood during session III (cf. Methods for details). **b** Proportion of test block choices between images that had been encountered without feedback during preceding reconsolidation in different blocks, i.e., pre-WoF or post-WoF. This analysis involved choices between images with objectively similar reward probabilities, i.e., high vs. high and low vs. low, and preference is computed as difference between percent choices of post-WoF images and pre-WoF images, as a function of WoF outcome. Contrasting a mood bias on learning, subjects preferred images that re-appeared post-WoF, irrespective of whether the WoF was won or lost. However, this effect was differentially modulated by the drug (drug × valence: $F_{1,60} = 4.8$, $p = 0.032$, $n = 64$), with a significant preference for images encountered after winning $F_{1,15} = 6.4$, $p = 0.023$) but not losing the draw in the SSRI group, and a preference for images encountered after losing ($F_{1,15} = 10.1$, $p = 0.006$) but not winning the draw in the placebo group. ** $p < 0.01$, * $p < 0.05$. Error bars indicate SEM.

manipulation or other task-related events. Importantly, there was no baseline difference in mood ratings in the pre-drug session ($F_{1,62} = 0.2$, $p = 0.633$). We found mood self-report did not differ between groups during treatment (drug: $F_{1,60} = 0.2$, $p = 0.616$; drug × session: $F_{2,120} = 1.2$, $p = 0.299$; Fig. 5a). Additionally, subjects provided multiple affective state questionnaires repeatedly over the course of the study (Beck's Depression Inventory (BDI-II[25]), Snaith−Hamilton Pleasure Scale (SHAPS[26]), State-Trait Anxiety Inventory (STAI[27]), and the Positive and Negative Affective Scale (PANAS[28]). We found no evidence for drug effects on any of these self-report questionnaires (Supplementary Table 1). This is in line with previous studies, in both healthy volunteers[7,13] and depressed patients[22], showing week-long SSRI treatment does not enhance mood directly.

Second, SSRIs could enhance positive affective responses to outcomes, i.e., mood responsivity, an effect that should be reflected in how subjects' mood changed immediately following the WoF draw, i.e., in block 2 compared to block 1. As described above, winning and losing the WoF draw had a distinct impact on affective state. However, mood responsivity was unaffected by SSRI treatment (drug: $F_{1,62} = 0.8$, $p = 0.361$; Fig. 5b), meaning that a positive ($F_{1,62} = 0.5$, $p = 0.482$) or a negative mood response ($F_{1,62} = 0.4$, $p = 0.499$) did not differ between groups. Thus, SSRI treatment did not modulate immediate mood responsivity.

Ultimately, instead of an immediate impact on affective responses, the affect-learning interaction account of SSRIs suggests they have a delayed positive impact on mood. Specifically, by enhancing sensitivity to rewards following a positive mood manipulation, SSRIs could increase the positive impact that these subsequent rewards have on mood. In our experiment, such impact of the post-WoF rewards should

manifest most clearly in the test block, after all block 2 rewards had been awarded. To test for this delayed impact of the WoF manipulation, we examined mood during the test block compared to mood during block 1.

Indeed, we found a distal WoF-impact was differentially modulated by drug ($F_{1,62} = 5.6$, $p = 0.020$). Note that this effect was not attributable to a significant drug effect for the win ($F_{1,62} = 1.7$, $p = 0.189$), or for the loss condition ($F_{1,62} = 0.28$, $p = 0.095$), but driven by an interaction of an amplified positive, and lower negative mood response in the SSRI as compared to the placebo group (Fig. 5c).

Note that we did not find a significant three-way interaction (factors 'time': 1st/2nd/3rd block, 'drug', 'valence'; $F_{2,122} = 1.8$, $p = 0.172$). This does not conflict with an effect of SSRIs on mood per se, but merely indicates the delayed drug effect was not significantly greater than the immediate effect. However, the finding of a delayed effect needs to be interpreted with caution.

Based on our hypothesis, a greater positive mood effect should be evoked by enhanced affect-learning dynamics. To test this prediction, we assessed the relationship between the positive mood bias parameter that modulated post-WoF learning (derived from the computational model) and the delayed mood response (assessed during test block). Strikingly, subjects with a high positive mood bias parameter showed a highly significant positive mood response ($F_{1,30} = 14.6$, $p < 0.001$), an effect not present in subjects with a low parameter estimate ($F_{1,30} = 0.8$, $p = 0.363$; high and low defined using a median split across the whole sample). Critically, this distal mood response was significantly different between high and low mood bias groups ($F_{1,60} = 9.9$, $p = 0.002$; Fig. 5d), and this delayed effect remained significant when controlling for the immediate affective response to the WoF ($F_{1,59} = 9.0$, $p = 0.004$). In a correlation analysis, across the entire

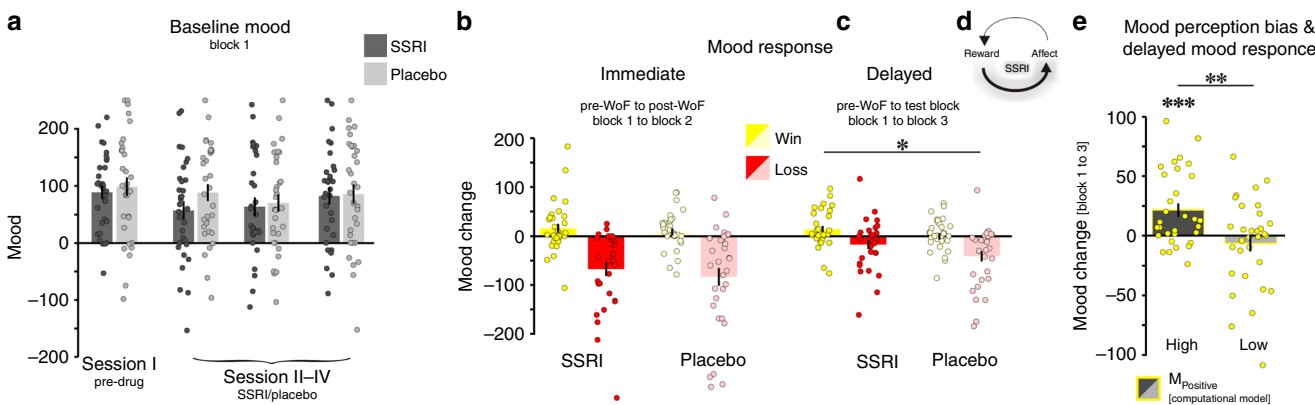

**Fig. 5 SSRIs induce mood effects via affect-learning interaction. a** Mean mood ratings during the first block of every session, i.e., prior to any mood manipulation. Mood self-reports did not differ between groups in the pre-drug session (session I), or during the 1-week drug period (sessions II−IV). Note that there were similarly no effects on affective state questionnaires, as repeatedly assessed over the course of the study (cf. Supplementary Table 1). **b** Immediate mood responsivity, as a function of WoF outcome, i.e., mean mood change from block 1 to block 2. WoF outcome profoundly affected emotional state, such that subjects reported to feel happier in the post-WoF block (block 2) after a win and less happy after a loss. However, immediate mood responsivity was unaffected by SSRI treatment. **c** Delayed mood responsivity, as a function of WoF outcome, i.e., mean mood change from block 1 to block 3. Subjects showed an ongoing mood response during test block (block 3), with elevated mood after a winning draw, and decreased mood after a losing draw. However, delayed mood responsivity was modulated by treatment, such that there was a greater positive mood effect in the SSRI group, and a greater negative mood effect in the placebo group ($F_{1,62} = 5.6$, $p = 0.020$, $n = 64$). **d** Interaction between affect and reward sensitivity. Rewards impact on emotional state, which, in turn, impacts on perception of subsequent rewards. Biased perception of reward, a downstream effect of a serotonergic impact on affect-learning dynamics (cf. Fig. 3b, c), can result in a delayed mood impact of the WoF that was only evident after subjects had played an additional block in which they received more rewards. **e** Relationship between positive mood perception bias and delayed positive mood response. Subjects with a high positive mood bias parameter, derived from the computational model, showed a strong delayed positive mood response ($F_{1,30} = 14.6$, $p < 0.001$), an effect not present in subjects with a low mood bias (division using a median split across the whole sample), and that significantly differed between the two groups ($F_{1,60} = 9.9$, $p = 0.002$, $n = 62$). Note that this positive mood bias parameter acts via boosting subjectively perceived reward during learning after winning the WoF, an effect significantly enhanced in the SSRI as compared to the placebo group. \*\*\*$p < 0.001$, \*\*$p < 0.01$, \*$p < 0.05$, n.s. = not significant (no difference across drug groups). Error bars indicate SEM.

sample, we found a similar significant relationship between a greater positive mood bias parameter and a greater delayed positive mood change ($R = 0.278$, $p = 0.030$; Supplementary Fig. 10). Thus, a positive mood learning bias, reflected in an enhanced responsivity to reward after the experience of a successful WoF draw, was associated with a positive subsequent mood effect[29]. Overall, SSRIs did not influence affect directly, but caused a delayed mood response, putatively a downstream effect of altered responsivity to reward. Note, however, a downstream effect, as a function of biased learning, is only one possible interpretation that requires further investigation.

## Discussion

We show that serotonergic antidepressants boost the influence of positive affect during both learning of novel, and reconsolidation of previously learned, reward associations. Thus, rather than impacting on affect and reward sensitivity directly, SSRIs appear to modulate a bilateral interaction between the two. We suggest such positive feedback dynamics give rise to an evolving incremental mood improvement, as well as provide a candidate mechanistic account for the unfolding of antidepressant effects over time.

Emotions and moods are considered interacting processes expressed over different timescales[30]. Reward induces positive emotion, and repeated exposure to rewards elicits enduring positive emotional states, i.e., elevated mood[31]. Here, we replicate previous findings showing that unexpected outcomes impact on emotional state, which, in turn, results in subsequent outcomes being perceived as better in a positive, and worse in a negative state[20,21]. This suggests a bidirectional relationship between affect and reward sensitivity, whereby a reward induces positive affect, and positive affect in turn renders subsequent rewards more valuable. Our novel finding is to show that serotonin boosts the impact of positive affect on reward sensitivity. SSRIs specifically enhance one axis of a two-way interaction, potentially engendering a positive feedback loop, whereby a boosted perception of positively biased rewards leads to further mood elevation[19]. Such non-linear dynamics may result in the gradual co-evolution of positive mood and a preponderance of positive experiences, thereby giving rise to the delayed clinical onset of SSRI therapy[22].

Our study extends an understanding of serotonin's role in the treatment of affective disorders at multiple levels. First, previous research suggests serotonergic antidepressants induce a bias in the perception of inherently emotional information, for example emotional face perception[15,16]. We show that SSRI-induced affective biases influence cognitive processes other than perception, such as learning and reconsolidation. Critically, mood disorders are often characterised by aberrant sensitivity to, and an inability to accurately learn from, reward[32,33]. Our study is a step towards understanding how antidepressants impact upon dysfunctional cognition in depression, and how a relatively immediate serotonergic effect on implicit affect-cognition interactions can set the scene for a mood improvement that emerges over time[17]. Second, positive feedback dynamics not only constitute a putative mechanism of antidepressant drug action but may also play a role in the emergence of manic episodes and mood swings[34]. A persistence of such escalatory dynamics has been proposed to engender overly optimistic expectations and elated mood, particularly in individuals prone to mood instability[21]. This might explain a frequently observed phenomenon of treatment-induced affective switching, and mood cycle acceleration, associated with antidepressant medication in bipolar disorder[35].

Serotonin's role in reward processing, and its resulting modulatory impact on mood, remains a topic of debate[36–38]. Our findings support influential theories of serotonin's role in reporting the affective value of the environment[39], and in facilitating an agent's ability to adapt flexibly to dynamic environments through plasticity and behavioural control[40]. Thus, the perception of large unexpected rewards may signal the current environmental context is beneficial, resulting not only in elevated mood but also in a positively biased perception of subsequent experiences[41]. This chimes with nonhuman animal studies of learning, which show antidepressants act to enhance an influence of affective state on subsequent reward valuation[42,43]. It remains to be demonstrated whether SSRIs amplify affect-learning interactions via an increase in tonic firing, phasic release of serotonin, or through interaction with other neuromodulators such as dopamine[38].

A limitation of our study is that we did not assess patients with clinical depression. Previous research indicates depressed individuals respond to experimental induction of positive mood[44,45], albeit affective response is typically attenuated and maintained for a shorter time period[45,46]. Critically, variability in the ability to experience positive affect predicts response to antidepressant treatment[47,48], and evidence indicates emotional reactivity can vary as a function of depression subtype[46,49]. Precisely how a serotonergic impact on affect-learning dynamics relates to treatment response will need to be targeted in future clinical studies. Another limitation pertains to the mechanistic understanding of a secondary finding, where positive and negative affect boosted subjective value during post-WoF reconsolidation. Although our study provides some evidence that the effects of mood on learning and reconsolidation constitute distinct processes, this question remains an important issue for future work aimed at unpacking whether an WoF effect on reconsolidation is indeed mediated by arousal. Moreover, we acknowledge that week-long SSRI treatment does not typically exert a clinically meaningful impact on depressive symptoms[22]. Longitudinal study designs, over longer timescales, that probe whether interindividual variability in candidate computational parameters (e.g., the influence of positive affect on learning) relates to clinical treatment response are an avenue for future research. Thus, studies combining pharmacological intervention, repeated behavioural testing, and ecological momentary mood assessment, might reveal whether an early serotonergic impact on affect-learning dynamics (within hours or days) provides a useful predictive marker for clinical response (after multiple weeks)[14,50,51].

In conclusion, we show that SSRIs boost positive affective biases in reward processing, an effect that may result in a delayed mood response. Such SSRI-induced affect-learning dynamics provide a framework for understanding the complex dynamics of mood and hold great promise as an early marker of treatment response that can aid tailoring of currently available, and development of novel, therapeutic approaches to depression.

## Methods

**Subjects.** Sixty-four healthy volunteers (mean age: 24.7 ± 4.0; range 20−38 years; 38 females; Supplementary Table 1) participated in this double-blind, placebo-controlled, mixed-factorial design study. Before study enrolment, subjects were medically screened over the phone, and assessed by a medical doctor at the study location upon arrival for the first study session. This assessment included an electrocardiogram to exclude QT interval prolongation, as well as a structured medical history, to ensure subjects had no current health issue, no history of neurological or psychiatric disorder, no prior allergic reaction to any drug, and were not taking any regular medication. The experimental protocol was approved by the University College London (UCL) local research ethics committee (Ethics-ID: 9787/002), with informed consent obtained from all participants.

**Pharmacological procedure.** Participants were randomly allocated to two equal-sized groups, controlling for an equal gender balance across both groups, to receive a daily oral dose of the SSRI citalopram (20 mg) or placebo, over a period of 7 consecutive days. Subjects were informed that there was a 50% chance that they

would be in the active drug group, and a 50% that they would be allocated to the placebo group. Additionally, subjects were informed regarding the double-blind procedure of the study, meaning that they were not told which group they had been allocated to, and that the study team responsible for data collection was also unaware about their drug group allocation. All subjects performed four laboratory testing sessions comprising a baseline session before start of the drug regime on day 1 (pre-drug) as well as three sessions that spanned the drug administration period, each session separated by three days, i.e., on days 1, 4, and 7 (Fig. 1d).

The first medication was taken at the study location at completion of the pre-drug session. Subsequently, all subjects returned for session II after 3 h, as citalopram reaches its highest plasma levels after this interval[52]. On the following days, subjects were asked to take their daily medication dose at a similar time of day, either at home or at the study location. All sessions occurred at a similar time of day, with tablets on study days being taken 3 h before test. The last session was on day 7 of treatment, a time when citalopram is known to reach steady-state plasma levels[53].

**Affective state questionnaires**. To examine putative effects of the drug on subjective affective states over the course of the study, participants completed the Beck's Depression Inventory (BDI-II[25]), Snaith−Hamilton Pleasure Scale (SHAPS[26]), State-Trait Anxiety Inventory (STAI[27]), and the Positive and Negative Affective Scale (PANAS[28]) on two different occasions: (i) pre-drug, session I, day 1; (ii) peak drug, session IV, day 7. Additional PANAS questionnaires, and another set of questionnaires not analysed as part of this study, were administered during breaks before the test block on each session in order to distract from the task and screen and prolong the break before the test block.

**Experimental task**. The computerised task was implemented using the Cogent 2000 Toolbox (http://www.vislab.ucl.ac.uk/cogent_2000.php) in MATLAB 2010b (Mathworks). Note that the visual stimuli in Figs. 1, 3a, 4a constitute representative stand-in images. The actual stimuli used in the experiment are available from the corresponding author upon reasonable request.

Learning blocks: To assess reward learning, subjects were tasked on each session with two separate blocks of a trial-and-error reward learning task (session I included two additional learning blocks, i.e., four in total, to familiarise subjects with the task). On each trial (Fig. 1a), subjects had 3 s to choose from one of two available images using left and right arrow keys on a computer keyboard. Once an image had been chosen, it was highlighted for 0.7 s by two white lines, followed by a 1.5 s display above the chosen image indicating either a reward (gold coin, £0.03) or no reward (red cross) outcome. After a fixation cross inter-trial interval of 2 s, the start of the next trial was indicated by the appearance of another pair of images.

Each block comprised 48 such trials involving a set of four novel images, two with a fixed high reward (70%), and two with a fixed low reward probability (30%). These probabilities were never revealed to subjects, though they were instructed that each image was associated with a fixed probability of reward and that to perform well and maximise monetary outcome they needed to choose images they considered more likely to lead to reward. To examine putative changes in subjects' learning throughout the week, every image appeared with reward feedback during one learning block alone. The two learning blocks were separated by a short self-paced break. Please see Supplementary Fig. 1 for details on session structure and image appearance across the entire study.

Test blocks: After the two learning blocks, subjects performed a test choice block that involved presentation of all images encountered during preceding learning, consisting of trials comparing images from block 1 vs. block 2. To prevent new learning at this stage, outcomes were not revealed. Instead, they were 'hidden in a box' visible to subjects (Fig. 1b). Subjects were informed that in making choices they should rely on what they had learned previously and that they would be rewarded for their choices as before. This test block comprised 48 trials, 24 trials involved pairs of images with different reward probabilities, i.e., high vs. low, and 24 trials with images of similar reward probabilities, i.e., high vs. high, and low vs. low, respectively.

Note that the session IV test block comprised 72 trials, containing 24 additional trials, on which subjects were asked to choose between images of similar reward probability, i.e., high vs. high, and low vs. low. Critically, this set of trials comprised images that were learned in the same block on the previous session, i.e., session III, but encountered as interleaved maintenance trials either before or after the WoF draw on the current session, i.e., session IV. This design feature allowed us to assess the impact of mood on reconsolidation of reward associations, and is described in detail below.

Affective state self-reports: During the task, subjects were asked to report their current emotional state by responding to the question 'How do you feel right now?' (Fig. 1c). The requested rating was provided without time restriction by moving a cursor (always starting at the midpoint) along a scale that spanned 'very unhappy' on the extreme left through to 'very happy' on the extreme right of the screen. The question appeared multiple times per block in fixed intervals (three times during blocks of 48 trials, four times during blocks of 72 trials).

Mood manipulation: To generate a large prediction error that might impact upon a subject's affective state, we used a WoF draw in breaks between learning blocks in session II (day 1) and IV (day 7; Fig. 3a). Possible outcomes, a win or loss of £1−£9, were depicted on the WoF, which gradually turned for approx. 45 s.

When the wheel stopped, an indicator above it revealed the outcome of the draw. Unbeknownst to subjects, and to avoid unwarranted differences in payment between subjects, the WoF draw was arranged such that each participant won and lost once, with the order counterbalanced across subjects. The actual amounts, randomly varying from £7 to £9, were sufficiently high to elicit an emotional response (cf. Results). Participants were notified in advance that they would be paid according to their earnings in the whole experiment, so all wins and losses were real.

Interleaved maintenance test trials during learning blocks: Subjects were instructed that they would encounter all images again in future sessions and would be asked to choose between images they had learned about on previous sessions. Similar to the test block, outcomes were not revealed on such trials. Crucially, images only re-appeared in the following session, i.e., images learned about in session I re-appeared in session II, images from session II in session III, etc. Such trials involving images from previous sessions were interleaved during learning blocks, with each block containing 24 such trials without feedback.

**Data analysis**. The statistical analysis was conducted using MATLAB 2014a (Mathworks) and SPSS 22 (IBM).

Affective state (days 1, 4 and 7): We assessed putative effects of the drug by computing change of affective state questionnaire scores from day 7 (session IV), i.e., at end of treatment period, vs. pre-drug (session I). Additionally, we assessed affective state self-reports measured during the experimental task. Here, baseline ratings for sessions I−IV were computed as mean mood rating during the first learning block of each session.

Mood manipulation (days 1 and 7): To assess the effect of the WoF on emotional state, we compared mean mood self-reports in pre- and post-WoF blocks during sessions II and IV.

In addition to an immediate impact on affective responses, we also assessed a delayed mood effect of the WoF. To test for this distal impact of the WoF manipulation, we examined mood during the test block compared to mood during block 1.

Learning & maintenance of reward associations (days 1, 4 and 7): In each test block, half the trials involved image pairs with different reward probabilities, i.e., high vs. low. Subjects' task performance on each day was quantified as proportion of high reward probability image choices on such trials. Data from sessions, in which subjects did not choose the high reward probability stimulus on >50% of high vs. low reward probability trials, were discarded from further analysis (two placebo/one SSRI session, i.e., 1.1% of total data).

Since all images re-appeared on interleaved maintenance trials, without feedback, during the subsequent session, subjects needed to retain learned reward associations to perform well. Here, performance was similarly quantified as proportion of high probability images choices on trials comparing images with different reward probabilities. This procedure allowed assessment of performance decline due to a temporal lag between initial learning and re-appearance.

Mood bias on learning (days 1 and 7): Based on our previous work[21], we predicted the WoF draw would impact not only on subjects' emotional state, but also on their subsequent valuations of rewards. To quantify such an interaction between emotional state and learning, we computed the proportion of chosen images encountered before (pre-WoF) vs. after (post-WoF) mood manipulation. Crucially, this analysis was conducted for test block trials that included image pairs with objectively similar reward probabilities that had been encountered in different blocks. There was no significant difference for (high vs. high) vs. (low vs. low) for the win ($F_{1,60} = 0.9$, $p = 0.350$) or loss ($F_{1,60} = 3.2$, $p = 0.086$) conditions. Thus, we collapsed trials, i.e., high vs. high, and low vs. low, to increase statistical power for the analysis of trials comprising images pairs with similar reward probabilities. We conducted the same analysis from the mood-neutral session taken place during drug treatment but without mood manipulation, i.e., session III. This result was regressed out from the analysis of our mood manipulations to control for putative primacy or recency effects irrespective of the WoF draw.

Mood bias on learning, logistic regression (days 1 and 7): In addition to directly comparing between images with equal reward probabilities, we conducted a more complete and individualised analysis of subjects' test choices, including choices between images with both equal and different reward probabilities. For this, we used a logistic regression, on each session that comprised a WoF draw, with choice on test block trials (without feedback) as the dependent variable on each session. As predictor variables, we used the difference in number of rewards received (during preceding learning with feedback) for the two images shown on the screen, and the block during which images were learned, i.e. pre- or post-WoF;

$$p(c_t = 2) = \frac{1}{1 + e^{-\left(\beta_0 + \beta_1 x_{1_t} + \beta_2 x_{2_t}\right)}},$$

where either the left ($c_t = 1$) or right ($c_t = 2$) image shown on the screen is chosen on trial $t$, $x_{1_t}$ is the difference in the number of experienced rewards during preceding learning for the right vs. the left image (e.g., 9 − 3 = 6), $x_{2_t}$ indicates whether the right image was learned in a post-WoF block (coded as the WoF outcome for images learned post-WoF, and 0 for images learned pre-WoF). Note that the test block only involved trials comparing images from block 1 vs. block 2. Thus, the right image was always encountered during a different learning block than the left image. Positive coefficients for the first predictor ($\beta_1$) indicate how

strongly subjects favoured images for which they had received more rewards during learning. Positive coefficients for the second predictor ($\beta_{2_{\text{positive}}}$, or $\beta_{2_{\text{negative}}}$, respectively) indicate how strongly subjects favoured images about which they had learned following a WoF win, and disfavoured images about which they had learned following a WoF loss. $\beta_0$ is the intercept.

Control analysis stimulus sampling, logistic regression (days 1, 4 and 7): To rule out a possibility that our results were affected by uncertainty aversion, we computed a logistic regression analysis, across all sessions, assessing whether the number of times a stimulus had been chosen during learning affected how often subjects chose it during the test block;

$$p(c_t = 2) = \frac{1}{1 + e^{-(\beta_0 + \beta_1 x_{1_t} + \beta_2 x_{2_t})}},$$

where either the left ($c_t = 1$) or right ($c_t = 2$) image on the screen is chosen on trial $t$, $x_{1_t}$ is the difference in the number of experienced rewards during preceding learning, for the right vs. the left image (e.g., 9 – 3 = 6), $x_{2_t}$ is the difference in the number of times an image had been chosen during preceding learning, for the right vs. the left image (e.g., 12 – 9 = 3). Positive coefficients for the first predictor ($\beta_1$) indicate how strongly subjects favoured images for which they had received more rewards during learning. Positive coefficients for the second predictor ($\beta_2$) indicate how strongly subjects favoured images they had chosen more often during learning (over and above the number of experienced rewards per image). $\beta_0$ is the intercept.

Mood bias on reconsolidation (day 7): To examine an effect of the WoF draw on reconsolidation of previously learned reward associations, subsets of images re-appeared in future sessions in a specific manner. For example, in session III, subjects learned about four novel images during both block 1 (set A) and 2 (set B), respectively. During session IV, half of these images re-appeared prior to (subsets A1 and B1) or subsequent to (subsets A2 and B2) the mood manipulation (without reward feedback) such that subjects had to rely on their learned associations from 3 days earlier. During the test block on session IV (Fig. 4b), subjects were exposed to an additional set of choices, consisting of 24 trials with images of objectively similar reward probabilities (e.g., A1 vs. A2, B1 vs. B2). By design, those images were learned within the same block, during a session without a mood manipulation 3 days earlier. However, they were encountered without feedback either before or after a positive or negative mood manipulation during the following session. This allowed us to quantify the effect of the WoF draw on reconsolidation by computing the proportion of chosen images that re-appeared before (pre-WoF) or after (post-WoF) the draw. Crucially, images always re-appeared in this fashion during the following sessions so as to facilitate comparability. However, this procedure was introduced specifically to test for the impact of mood on reconsolidation, i.e., for images encountered initially during session III and re-appearing during session IV. This analysis comprised test block trials comparing image pairs with objectively similar reward probabilities that had been re-encountered in different blocks. There was no significant difference for (high vs. high) vs. (low vs. low) for the win ($F_{1,62}$ = 0.06, $p = 0.801$) or loss ($F_{1,62}$ = 0.3, $p = 0.583$) conditions. Thus, we collapsed trials, i.e., high vs. high, and low vs. low, to increase statistical power for the analysis of trials comprising images pairs with similar reward probabilities.

**Computational modelling**. Modelling: learning from reward: To identify the computations that guided subjects' behaviour, we compared a set of computational learning models in terms of how well each model fitted subjects' choices.

First, we determined how subjects learned from outcomes through trial-and-error. We used standard reinforcement learning models, which learn the expected value ($Q$) of an image by updating it according to a reward prediction error ($\delta$), reflecting the difference between obtained outcome ($R$) and expected outcome ($Q$):

$$\delta_t = R_t - Q_t(s_{t,c_t}),$$

where $s_{t,c_t}$ is the image chosen at trial $t$, $Q_t(s)$ is the expected outcome for image $s$ at trial $t$, $R_t$ is the outcome at trial $t$ ($R = 0$ corresponds to a no-reward, and $R = 1$ corresponds to a reward outcome), and $\delta_t$ is the prediction error at trial $t$.

Model 1 has a fixed learning rate parameter ($\eta$) that weights the influence of prediction errors on subsequent expectations and assigns greater weight to more recent outcomes:

$$Q_{t+1}(s_{t,c_t}) = Q_t(s_{t,c_t}) + \eta \delta_t.$$

In contrast, Model 2 has a dynamic learning rate ($\alpha$) that changes as a function of the number of observed outcomes ($N$) for the chosen image. Here, $\varepsilon > 0$ is a free parameter that determines the initial learning rate. The learning rate gradually decreases asymptotically toward zero with the number of observed outcomes ($N$):

$$Q_{t+1}(s_{t,c_t}) = Q_t(s_{t,c_t}) + \alpha_t \delta_t,$$
$$\alpha_t = \frac{1}{\varepsilon + N_t(s_{t,c_t})}.$$

Such a decrease in learning rate is consistent with optimal inference given fixed reward probabilities[54]. In both models, a Softmax function is used to derive choice probabilities ($p$) according to learned expectations. The models then choose an image (either the left image $c_t = 1$, or the right image $c_t = 2$), with an inverse temperature parameter ($\beta$) determining the strength with which choices are directed towards higher value options:

$$p(c_t = i) = \frac{e^{\beta} Q_t(s_{t,i})}{\sum_{i=1}^{2} e^{\beta} Q_t(s_{t,i})}.$$

Modelling: mood biasing reward perception: Based on earlier work[19], we hypothesised mood to bias subjective value during learning. To account for effects of mood on valuation, models 3 (fixed learning) and 4 (dynamic learning) compute prediction errors with respect to perceived rather than actual reward:

$$\delta_t = R_{\text{perceived}_t} - Q_t(s_{t,c_t}).$$

Here, perceived reward ($R_{\text{perceived}}$) is different from actual reward ($R$) in that it reflects the biasing effect of mood ($M$). Such an additive bias is consistent with a normative role for mood in accounting for momentum in reward changes[19]. After winning the WoF draw, this bias was computed as:

$$R_{\text{perceived}_t} = R_t + M,$$

whereas the opposite bias was implemented after losing the draw:

$$R_{\text{perceived}_t} = R_t - M.$$

Here, $M = 0$ does not bias perception of reward, whereas $M > 0$ exerts positive feedback as reward is perceived as larger in a good mood and as smaller in a bad mood (and vice versa for $M < 0$).

Notably, this model assumes that the effect of positive and negative mood is qualitatively different, but quantitatively similar. Thus, we next tested whether positive and negative mood exerted distinct effects across participants. Here, models 5 (fixed learning) and 6 (dynamic learning) incorporate different parameters for positive ($M_{\text{positive}}$) and negative ($M_{\text{negative}}$) mood after winning:

$$R_{\text{perceived}_t} = R_t + M_{\text{positive}},$$

and losing the WoF draw:

$$R_{\text{perceived}_t} = R_t + M_{\text{negative}}.$$

Modelling: mood biasing rate of learning: Additionally, we hypothesised that, instead of biasing reward perception, mood could bias the rate of learning from reward. Here, in models 7 (fixed learning) and 8 (dynamic learning), rate of learning reflects the biasing effect of mood ($M$), after winning the WoF draw:

$$\eta = \eta + M \,(\text{model } 7)$$

$$\alpha_t = \alpha_t + M \,(\text{model } 8),$$

and after losing the draw:

$$\eta = \eta - M \,(\text{model } 7)$$

$$\alpha_t = \alpha_t - M \,(\text{model } 8).$$

Finally, models 9 (fixed learning) and 10 (dynamic learning) incorporate different parameters for positive ($M_{\text{positive}}$) and negative ($M_{\text{negative}}$) mood after winning:

$$\eta = \eta + M_{\text{positive}} \,(\text{model } 9)$$

$$\alpha_t = \alpha_t + M_{\text{positive}} \,(\text{model } 10),$$

and losing the WoF draw:

$$\eta = \eta + M_{\text{negative}} \,(\text{model } 9)$$

$$\alpha_t = \alpha_t + M_{\text{negative}} \,(\text{model } 10).$$

Model fitting: To fit parameters of different models to subjects' decisions, we used an iterative hierarchical expectation-maximisation procedure across the entire sample[29,55]. We sampled $10^5$ random settings of the parameters from predefined prior distributions. Then, we computed the likelihood of observing subjects' choices given each setting and used the computed likelihoods as importance weights to re-fit the parameters of the prior distributions. These steps were repeated iteratively until model evidence ceased to increase. To derive the best-fitting parameters for each individual subject, we computed a weighted mean of the final batch of parameter settings, in which each setting was weighted by the likelihood it assigned to the individual subject's decisions. Note that the hierarchical fitting procedure, including all priors, were applied to the entire sample without distinguishing between SSRI and placebo subjects. This ensured that the parameter estimates, at the level of individual subjects, were mutually independent given the shared prior, rendering it appropriate to assess between-group differences.

Fixed learning rates ($\eta$) were modelled with Beta distributions (initialised with shape parameters $a = 1$ and $b = 1$). Mood parameters ($M$, $M_{\text{positive}}$, $M_{\text{negative}}$) were modelled with normal distributions to allow for both positive and negative effects (initialised with $\mu = 0$ and $\sigma = 1$). The remaining parameters ($\varepsilon$, $\beta$) were modelled with Gamma distributions (initialised with $\kappa = 1$, $\theta = 1$).

Model comparison: We compared between pairs of models in terms of how well each model accounted for subjects' choices by means of the integrated Bayesian

Information Criterion (iBIC[29,56]). Here, we estimated the evidence in favour of each model ($\lambda$) as the mean likelihood of the model given $10^5$ random parameter settings drawn from the fitted group-level priors. We then computed the iBIC by penalising the model evidence to account for model complexity as follows: iBIC = $-2 \ln \lambda + \kappa \ln n$, where $\kappa$ is the number of fitted parameters and $n$ is the total number of subject choices used to compute the likelihood. Lower iBIC values indicate a more parsimonious model fit. We validated this model comparison procedure by generating simulated data using each model and applying our model comparison procedure to recover the model that generated each dataset (cf. Supplementary Fig. 7).

**Reporting summary**. Further information on research design is available in the Nature Research Reporting Summary linked to this article.

## Data availability

The data that support the findings of this study are available from the corresponding author upon reasonable request.

## Code availability

The custom computer code used to generate results that are reported in the paper are available from the corresponding author upon reasonable request.

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

## Acknowledgements

We thank Robb B. Rutledge and Quentin J.M. Huys for helpful comments on a previous version of this manuscript. J.M. was supported by a fellowship from the German Research Foundation (MI 2158/1-1). R.J.D. holds a Wellcome Trust Investigator award (098362/Z/12/Z). The Max Planck UCL Centre for Computational Psychiatry and Ageing Research is a joint initiative supported by the Max Planck Society and University College London. The Wellcome Centre for Human Neuroimaging is supported by core funding from the Wellcome Trust (091593/Z/10/Z).

## Author contributions

J.M., E.E. and R.J.D. designed the experiments. J.M. and I.M.M. performed the experiments. J.M. and E.E. analysed and interpreted the data. J.M., E.E. and R.J.D. wrote the paper.

## Competing interests

The authors declare no competing interests.
