## [Peer Review File · Nature Communications]

Reviewers' Comments:

Reviewer #1:

Remarks to the Author:

This is a drug study in healthy humans using SSRIs. The authors conducted a reward learning task in subjects on SSRI or placebo in several sessions over a week of drug treatment.

I thought this was an interesting study with some intriguing if not entirely clear-cut results. I cannot find anything to object to in the methodology, analysis or model. I cannot really comment on the drug protocol as this is not my field. My comments therefore concern the interpretation of the results and propose one or two possible further analyses.

Learning results -

The key finding concerns the impact of mood on learning. During some sessions, high- or low- mood was induced by exposing subjects to a large win (or no win) on a wheel of fortune draw, that was separate from the actual task. Subjects on SSRI behaved as though they learned higher values for stimuli that they experienced in a high mood state than those they experienced in a low mood state. This was not the case for subjects on placebo. The authors present a formal model of this phenomenon as a modification of a Rescorla-Wagner type learning rule with positive and negative mood terms, essentially capturing the idea that positive mood makes rewards seem more valuable and negative mood makes rewards seem less valuable.

I did wonder whether, in addition to the effect of mood induction on learning, there was an overall preference for recently learned stimuli in the SSRI subjects (Fig 3b- average of yellow and red bars for SSRI subjects would be positive, showing an overall preference for post-WOF stimuli, which are always more recent than pre-WOF, for these subjects). I guess this could be tested using the data from day 4, on which no mood induction was used – do subjects still prefer stimuli encountered in the second learning block? If so they have a preference for recently learned stimuli.

If such a preference were found, a rational explanation for it would be that subjects have lower uncertainty about the value of recently learned stimuli, and all else being equal (which it was in the test block), subjects prefer stimuli about which they have lower uncertainty. And that SSRI subjects are either more uncertainty averse. That possibility could be explored by asking whether subjects (or, especially, SSRI subjects) prefer stimuli that they sampled more times during learning (there should be some variation in this since subjects freely sampled) – over and above effects of learned stimulus value.

Reconsolidation results -

A secondary finding of the study was that if participants were reminded of stimuli (learned on a previous day) directly after the mood induction manipulation, their later preference for these stimuli depended on an interaction of mood and drug. SSRI subjects preferred stimuli of which they were reminded in a high mood state, whereas placebo subjects preferred stimuli of which they were reminded in a low mood state.

The possible mechanism for this secondary finding – the effect of the mood induction on preferences for previously learned stimuli (figure 4b) – is less well explained than the key result (learning/ fig 3b). At the bottom of p11 the authors suggest that positive mood induction particularly affects

reconsolidation in SSRI subjects but I am not clear what this means. What does It mean to say that the stimuli encountered in the positive mood state we better reconsolidated (for SSRI subjects)? Does it mean that those stimuli were remembered more accurately (but this should not bias choice towards those stimuli, which is what the data suggst), or that their value increased by re-encountering them in the positive mood state? The latter is compatible with the data for SSRI subjects but I can't clearly see what the parallel argument would be, to explain why placebo subjects seemed to prefer images reconsolidated in a low mood state...?

Reviewer #2:

Remarks to the Author:

Summary of Main Findings

Overall, this is a well-written, well-grounded and rigorous paper. The authors found, behaviorally, that treatment condition differently modulates the effect of affect induction (WoF) on learning, such that only the SSRI group prefers items that were learned after a positive affect induction. In memory, the results suggest a more lasting memory for items encountered after the affect manipulation, and this effect was also modulated by treatment group. However, the interpretation of both results could be further developed, specifically, to consider the possibility of effects on loss aversion (see "1. substantial concerns"). The results also appear to show through computational modeling that there is a differential effect of SSRI on the m_{positive} parameter, yet it was unclear if this effect was dependent on model fitting procedures (e.g. the setting of predefined group priors). This should be clarified (see "2. substantial concerns"). Finally, although the modeling provides a potential mechanism for the delayed response of SSRI (positive feedback between affect and reward sensitivity), the case for bidirectionality could be clarified (see "3. substantial concerns"). These results are potentially impactful, however, the substantial concerns should be addressed before proceeding.

Substantial Concerns

1) Behavioral results in figure 3 and 4 need to be further developed. Possible alternative theories including loss aversion should be addressed.

2) Priors - Were the group distributions "predefined", as stated, or was expectation maximization used to find the group priors in a hierarchical fashion? If the former, this could bias the critical m -parameter results, so we need some evidence that the results are not sensitive to the choice of group priors, as well as clarification on where the group priors were drawn from. If the latter, were different priors fit to drug condition, or was it simply assumed that all subjects regardless of drug condition were drawn from a common distribution? If different priors were used per drug condition then any statistical test between the groups might be unduly influenced by the shrinkage imposed by the priors. Please provide more details on these methods and statistical tests used.

3) Bidirectionality

Bidirectionality should be diagrammed for clarity and how the data and model supports the arrow going in each direction.

I recommend that to fully argue the point of directionality, as well as how this study (time scale testing = day) can be extrapolated to the time scale of clinical effect of SSRIs (weeks), there needs to be more discussion how this positive feedback loop evolves in time. One clear prediction of the positive feedback loop would be that the as the days progress, the immediate response of the SSRI group would be significantly higher than the placebo.

The claims of bidirectionality and mechanism would be weakened without this pattern.

Minor Concerns

Generalization to clinical population: is there a reason to believe that this will generalize to the clinical population. Specifically, considering this mechanism involves a positive feedback - would the clinical population have the initial increased affect to allow for this mechanism.

Abstract

Sentence 3 I would suggest making a change such as adding "experimentally induced" before "positive affect on..." This would make more salient that the key effect is an interaction between induced affect change and learning/reconsolidation.

Sentence 4 I'm not sure why "resulting in a delayed mood response" is in this sentence about computational modeling, as it seems this is an implication of the results as a whole, including the rule-outs of direct effects on general affect or affect responsivity to outcomes, rather than something shown directly by the modeling (although it is supported by the median split in Fig. 5D). It may also be helpful to clarify after "subjective reward perception during learning" that this was specifically following a positive (but not negative) induced affect change, as I think this would lead to a clearer segue to the next sentence.

Introduction

p. 3 I found paragraph 2 somewhat hard to follow. It begins by talking about negative cognitive biases in attention and memory. Then it suggests CBT modulates such biases. I took this to mean CBT tries to decrease negative cognition (as, indeed is typically the primary focus in, eg, Beckian Cognitive Therapy). However, it then states pharmacological intervention may work similarly, in that it induces a "rosier perspective" by boosting how positive information is perceived; it's unclear what this has to do with negative biases. I would recommend reworking this paragraph to make the flow clearer.

One option would be to cite literature that finds decreased attention to, and interpretation in the direction of, positive information in depression (eg, Armstrong & Olantunji, 2012; Evereart, Podina, & Koster, 2017). And then to note that a target of many CBTs for depression, such as Behavioral Activation, is to increase contact with rewarding information. This I think would be a clearer segue to discussing how pharmacological treatment may similarly induce a rosier perspective.

p. 3 It's unclear in what sense reward learning or memory consolidation is a "higher cognitive process" than what the authors are referring to as affective biases; consider rewording.

p. 9 I'm unclear on what the authors did in the logistic regression analysis. Please try to explain what's being tested more clearly. I think it would help to put a "such that..." after the effect of mood bias across the whole sample. Also, in Supp. Fig 1., the (B) should be placed before the text stating the effect.

Methods

It would be helpful to include in "Data Analysis" information about the regression analyses performed, as currently it's unclear what statistics the authors actually ran.

p. 18 Need more information about the "thorough medical screening to exclude any neurological or psychiatric interview." Who performed screening and what tools (eg, structured interview) were used?

When was the screening given?

It would be helpful to explain briefly what subjects were told re: the possibility of getting SSRI or placebo.

p. 20 The sentence beginning "Note that the session IV test block..." seems like it should go in the preceding section, "Learning blocks."

p. 23 Sentence beginning "Since all images re-appeared.." I'm having difficulty following what this is referring to—is it the interleaved maintenance trials? It would be helpful to clarify.

Results

Figure 3 title: "SSRIs boost the impact of positive mood on learning" - this implies that there is some baseline impact of positive mood on learning - however based on the fact that there is no preference in the placebo group when there is a win, it seems that this baseline does not exist.

Discussion

This theory seems to make precise experimental predictions. If the authors have space, I'd recommend noting some of these as a guide for future research about how their claims could be tested / falsified. For example, the theory seems to predict that as depressed individuals start an SSRI, event-contingent increases in good mood should lengthen gradually over time, mediated by intervening positive learning experiences. Moreover, how much good moods are lengthened should be moderated by individual differences in the M_{positive} parameter. The latter could be tested by combining computational modeling and EMA, and the former could be tested just using EMA.

Reviewer #3:

Remarks to the Author:

The authors demonstrated week-long SSRI treatment enhanced the impact of positive affect on learning of novel, and reconsolidation of previously learned, reward associations, by a placebo-controlled study.. Based on the result, they concluded SSRIs amplify a bilateral interaction between affect and reward sensitivity instead of influencing them directly.

The main research findings of this paper will be quite important for understanding of serotonin's delayed antidepressant effects.

However, I have to express several concerns as below.

Major points

I think the authors combined the 70%vs70% and 30% vs30% tests.

Didn't author analyze 70%vs70% and 30% vs30% tests separately?

In my opinion, 70% and 30% images themselves might contain positive and negative valence, respectively.

In figure 4B (page 12), subjects preferred images encountered after losing but not winning the draw in the placebo group. The fact might not correspond to the fact described in figure 3B; More detailed explanations from the perspective of reward associations would be needed. Why did participants prefer post-WoF after loss?

They tried to interpret from the perspective of "arousal", which is quite ambiguous.

Minor points

In my opinion the description of the methodology and technical details should be shortened considerably. The experimental procedures of the scheme could be summarized in a table instead of being described in the text.

The introduction and discussion sections seem to include several repetitions which could be shortened. In discussion part, it seems inadequate to clearly conclude that these findings fill a previously missing link (page 16, line9-11), only from written results. Seven days of research might not be enough to generalize the theory as the authors mentioned in the limitation part. What "an early marker" in the last sentence of the discussion part refers to is unclear.

We thank the reviewers for the positive evaluation of our manuscript “A mechanistic account of serotonin’s impact on mood” (NCOMMS-19-1125295). We are grateful for the opportunity to revise the manuscript and believe the revised version is substantially improved in light of the reviewers’ comments.

We addressed all comments point-by-point. The respective changes in the manuscript are marked in bold. Please note that the page numbers given below refer to the page numbers of the revised manuscript.

Reviewer #1:

This is a drug study in healthy humans using SSRIs. The authors conducted a reward learning task in subjects on SSRI or placebo in several sessions over a week of drug treatment.

I thought this was an interesting study with some intriguing if not entirely clear-cut results. I cannot find anything to object to in the methodology, analysis or model. I cannot really comment on the drug protocol as this is not my field. My comments therefore concern the interpretation fo the results and propose one or two possible further analyses.

We thank the reviewer for a positive evaluation of our study and the constructive suggestions. Below we address the reviewer’s proposed suggestions.

1) Learning results

The key finding concerns the impact of mood on learning. During some sessions, high- or low- mood was induced by exposing subjects to a large win (or no win) on a wheel of fortune draw, that was separate from the actual task. Subjects on SSRI behaved as though they learned higher values for stimuli that they experienced in a high mood state than those they experienced in a low mood state. This was not the case for subjects on placebo. The authors present a formal model of this phenomenon as a modification of a Rescorla-Wagner type learning rule with positive and negative mood terms, essentially capturing the idea that positive mood makes rewards seem more valuable and negative mood makes rewards seem less valuable.

I did wonder whether, in addition to the effect of mood induction on learning, there was an overall preference for recently learned stimuli in the SSRI subjects (Fig 3b- average of yellow and red bars for SSRI subjects would be positive, showing an overall preference for post-WOF stimuli, which are always more recent than pre-WOF, for these subjects). I guess this could be tested using the data from day 4, on which no mood induction was used – do subjects still prefer stimuli encountered in the second learning block? If so they have a preference for recently learned stimli.

If such a preference were found, rational explanation for it would be that subjects have lower uncertainty about the value of recently learned stimuli, and all else being equal (which it was in the test block), subjects prefer stimuli about which they have lower uncertainty. And that SSRI subjects are either more uncertainty averse. That possibility could be explored by asking whether subjects (or, especially, SSRI subjects) prefer stimuli that they sampled more times during learning (there should be some variation in this since subjects freely sampled) – over and above effects of learned stimulus value.

The reviewer is correct that the results depicted in Fig. 3B suggest a preference for stimuli learned in the post-WoF block in SSRI subjects. Moreover, the reviewer acknowledges a critical aspect of our study design – a mood-neutral session on day 4, i.e., session III, of drug treatment – identical to sessions II and IV, with the only difference that no WoF draw was held between learning blocks. This mood-neutral session allowed us to compute two specific control analyses that ensure the SSRI effects in our study are not confounded by putative primacy or recency effects.

Firstly, analysis of test block preference of the mood-neutral session (preference for block 1 or 2, independent of a WoF draw) revealed no difference between treatment groups ($F=0.5$, $P=0.470$). Secondly, we used this measure (i.e., the result from session III) as a covariate, to control for putative primacy or recency effects, in our mood bias analysis (i.e., the results from sessions II & IV). Reassuringly, mood-dependent drug effects remained significant when regressing out the effects from the mood-neutral session (win: $F=6.9$, $P=0.011$; loss: $F=0.6$, $P=0.807$; drug x valence: $F=4.8$, $P=0.032$).

In fact, we had reported this analysis in the previous version of the manuscript, but should have been more explicit as to its importance. We thank the reviewer for bringing this to our attention.

We now present the control analyses in more detail in the results section, and also added a paragraph emphasizing the rationale of the study design in the respective methods section of the revised manuscript.

“To rule out a possibility that results were driven by primacy (preference for block 1) or recency (preference for block 2) effects, we assessed test block choices from the mood-neutral session III (preference for block 1 or 2, in the absence of a WoF draw). We found no difference between treatment groups ($F_{1,60}=0.5$, $P=0.470$). Secondly, mood-dependent drug effects from session II and IV remained significant when regressing out effects from the mood-neutral session III (win: $F_{1,59}=6.9$, $p=0.011$; loss: $F_{1,59}=0.6$, $p=0.807$; drug x valence: $F_{1,59}=4.8$, $p=0.032$), indicating that the reported findings are independent of primacy or recency effects.” (page 8, results)

“We conducted the same analysis from the mood-neutral session taken place during drug treatment but without mood manipulation, i.e., session III. This result was regressed out from the analysis of our mood manipulations to control for putative primacy or recency effects irrespective of the WoF draw.” (page 26, methods)

Relatedly, the reviewer raises another relevant issue, namely the putative influence of sampling during learning on subsequent test block preference. To address this, we conducted two additional relevant analyses: first, we found no difference between treatment groups in the sampling of high or low reward stimuli across sessions ($F=0.0$; $P=0.984$); second, we computed a logistic regression analysis assessing whether the number of times a stimulus had been chosen during learning affected how often subjects chose it during the test block;

$$p(c_t = 2) = \frac{1}{1 + e^{-(\beta_0 + \beta_1 x_{1t} + \beta_2 x_{2t})}}$$

, where either the left ($c_t=1$) or right ($c_t=2$) image on the screen is chosen on trial t , x_{1t} is the difference in the number of experienced rewards during preceding learning, for the right versus

the left image (e.g., $9 - 3 = 6$), x_{2t} is the difference in the number of times an image had been chosen during preceding learning, for the right versus the left image (e.g., $12 - 9 = 3$).

Over and above a strong effect for β_1 , as already reported in the manuscript, we found a small, yet significant effect for β_2 ($F=8.7$; $P=0.005$). However, this measure was unaffected by drug treatment ($F=0.2$; $P=0.692$). If anything, the group mean for the β_2 regression coefficient was slightly higher for placebo subjects (mean: 0.057) as compared to SSRI subjects (mean: 0.043), indicating that the effects we describe are not driven by uncertainty aversiveness in SSRI subjects.

We thank the reviewer for raising this important issue and present the two control analyses in the revised version of the manuscript.

“To rule out a possibility that our results were affected by how often subjects had sampled images during learning, we conducted two further control analyses. Firstly, we ruled out any difference between treatment groups in their sampling of images with high or low reward probability across sessions ($F_{1,62}=0.00$; $P=0.984$). Secondly, we computed a logistic regression analysis to assess whether the number of times an image had been chosen during learning affected how often subjects chose it subsequently during the test block. Here, we found a small, yet significant effect ($F_{1,62}=8.7$; $P=0.005$), indicating subjects preferred images which they had sampled more frequently during learning and therefore had higher certainty about (over and above the number of experienced rewards per image). Crucially, this measure was unaffected by drug treatment ($F_{1,62}=0.2$; $P=0.692$), indicating that SSRI treatment did not impact uncertainty aversion.” (page 10, results)

“Control analysis stimulus sampling, logistic regression [days 1 & 4 & 7]

To rule out a possibility that our results were affected by uncertainty aversion, we computed a logistic regression analysis, across all sessions, assessing whether the number of times a stimulus had been chosen during learning affected how often subjects chose it during the test block;

$$p(c_t = 2) = \frac{1}{1 + e^{-(\beta_0 + \beta_1 x_{1t} + \beta_2 x_{2t})}}$$

, where either the left ($c_t=1$) or right ($c_t=2$) image on the screen is chosen on trial t , x_{1t} is the difference in the number of experienced rewards during preceding learning, for the right versus the left image (e.g., $9 - 3 = 6$), x_{2t} is the difference in the number of times an image had been chosen during preceding learning, for the right versus the left image (e.g., $12 - 9 = 3$). Positive coefficients for the first predictor (β_1) indicate how strongly subjects favoured images for which they had received more rewards during learning. Positive coefficients for the second predictor (β_2) indicate how strongly subjects favoured images they had chosen more often during learning (over and above the number of experienced rewards per image). β_0 is the intercept.” (page 28, methods)

2) Reconsolidation results

A secondary finding of the study was that if participants were reminded of stimuli (learned on a previous day) directly after the mood induction manipulation, their later preference for these stimuli depended on an interaction of mood and drug. SSRI subjects

preferred stimuli of which they were reminded in a high mood state, whereas placebo subjects preferred stimuli of which they were reminded in a low mood state.

The possible mechanism for this secondary finding - the effect of the mood induction on preferences for previously learned stimuli (figure 4b) - is less well explained than the key result (learning/ fig 3b). At the bottom of p11 the authors suggest that positive mood induction particularly affects reconsolidation in SSRI subjects but I am not clear what this means. What does it mean to say that the stimuli encountered in the positive mood state were better reconsolidated (for SSRI subjects)? Does it mean that those stimuli were remembered more accurately (but this should not bias choice towards those stimuli, which is what the data suggest), or that their value increased by re-encountering them in the positive mood state? The latter is compatible with the data for SSRI subjects but I can't clearly see what the parallel argument would be, to explain why placebo subjects seemed to prefer images reconsolidated in a low mood state...?

We apologise for lack of clarity in the description of this aspect of our results. In terms of interpretation, we agree with the reviewer that our results are compatible with the WoF draw acting to increase the value of images re-encountered during post-WoF blocks, rather than subjects simply remembering those images more accurately.

In fact, our design allowed us to rule out the latter interpretation. When we assess whether performance (on high vs. low reward probability trials) differed between images re-encountered during the post-WoF and the pre-WoF block, we find no significant difference (post-WoF vs. pre-WoF: $F=1.2$, $P=0.302$). Moreover, we found no evidence that subjects performed more accurately as a function of valence of the draw (post-WoF win vs. post-WoF loss: $F=0.4$, $P=0.538$), or as a function of drug condition (post-WoF drug: $F=0.04$, $P=0.845$, post-WoF drug x valence: $F=0.7$, $P=0.417$). These results rule out an interpretation based upon images being remembered more accurately either as a function of mood induction or of drug condition.

We agree with the referee that the finding in SSRI subjects is compatible with a mood-congruent effect in the positive domain, such that positive mood boosts subjective value during both learning of novel, and reconsolidation of previously learned, reward associations.

The finding in placebo subjects is puzzling at first sight. However, interpretation here is aided by the fact that across the entire sample, there is a fundamental difference between the effect of WoF on learning and reconsolidation (cf. new supplementary figure below). Specifically, overall, subjects preferred images reconsolidated during the post-WoF block. Thus, in contrast to the effect on learning, this latter effect is consistent with a valence-independent boost in subjective value of post-WoF images, possibly mediated through the enhancement of arousal by the WoF draw (since, overall, this effect was evident following both positive and negative mood induction).

This difference between the two effects of the WoF, on learning and on reconsolidation, is further substantiated by the absence of any relationship between the two (win: $R=0.07$, $P=0.705$; loss: $R=-0.18$, $P=0.303$), suggesting the two constitute distinct processes.

Why both positive and negative affect would boost subjective value during reconsolidation is an important question for future work. Importantly, however, the impact of the WoF on both learning and reconsolidation were modulated by SSRI in the same way, which aligns with our predictions – namely, in both cases SSRIs boosted the impact of positive affect, biasing choices towards stimuli encountered in this state.

We agree with the reviewer the reconsolidation results deserve a more accurate description. We now add these new analyses and clarifications to the revised manuscript, including

presenting additional figures, depicting the results across the entire sample, in supplementary material.

“Across the entire sample, subjects preferred images that reappeared post-WoF, irrespective of whether they were linked to a win or loss (assessed on equal reward probability trials; $F_{1,60}=7.5$, $P=0.008$; Fig. 4B). This raises a possibility that increased arousal, engendered by the outcomes of the WoF draw, enhanced the subjective value of images during post-WoF reconsolidation, inducing a consequential preference for these images. This points to a possibility of a qualitatively distinct effect of the WoF draw on learning and on reconsolidation (Supplementary Fig. S2 for effects across the entire sample), a suggestion further supported by the absence of any relationship between the two effects (win: $R=0.07$, $p=0.705$; loss: $R=-0.18$, $P=0.303$).

Supplementary Figure 2. Effect of mood on learning and reconsolidation, across entire sample.

(A) Proportion of test block choices between images encountered during preceding *learning* in different blocks, i.e., pre-WoF or post-WoF. This analysis compared choices between images with equal reward probabilities, i.e., high vs. high and low vs. low, and preference is computed as difference between percent choices of post-WoF images and pre-WoF images, as a function of WoF outcome.

Participants showed a preference for images encountered in better mood, favouring post-WoF images after a winning draw, and favouring pre-WoF images after a losing draw.

(B) Proportion of test block choices between images that had been encountered without feedback during preceding *reconsolidation* in different blocks, i.e., pre-WoF or post-WoF. This analysis involved choices between images with objectively similar reward probabilities, i.e., high vs. high and low vs. low, and preference is computed as difference between percent choices of post-WoF images and pre-WoF images, as a function of WoF outcome.

Contrasting a mood bias on learning, subjects preferred images that re-appeared post-WoF, irrespective of whether the WoF was won or lost. ** $p<0.01$, * $p<0.05$, (*) $p=0.1$. Error bars indicate SEM.

Critically, an arousal bias was differentially modulated by treatment (drug \times valence: $F_{1,60}=4.8$, $P=0.032$; Fig. 4B), where the SSRI group showed a significant preference for images encountered after winning, but not after losing, (win: $F_{1,15}=6.4$, $P=0.023$; loss: $F_{1,15}=1.4$, $P=0.246$); and the placebo group after losing but not winning (win: $F_{1,15}=0.5$, $P=0.819$; loss: $F_{1,15}=10.1$, $P=0.006$).

We found no evidence that subjects performed more accurately (assessed on high vs. low probability trials), as a function of the draw (post-WoF vs. pre-WoF: $F_{1,62}=1.3$, $P=0.301$), valence of the draw (post-WoF win vs. post-WoF loss: $F_{1,60}=0.4$, $P=0.538$), or drug condition (post-WoF drug: $F_{1,60}=0.04$, $P=0.845$, post-WoF drug \times valence: $F_{1,60}=0.7$, $P=0.417$). These

observations rule out an explanation based upon images being remembered more accurately as a function of mood induction or of drug condition.

Thus, the findings are consistent with the WoF draw amplifying the subjective value of images re-encountered in the post-WoF block, leading to a biased choice preference towards such images. As predicted, SSRI treatment induced a preference for images re-encountered in a positive, compared to a negative, mood state.” (page 13/14, results; and supplementary material).

We have added further discussion of this result to the revised manuscript:

“Another limitation pertains to the mechanistic understanding of a secondary finding, where positive and negative affect boosted subjective value during post-WoF reconsolidation. Although our study provides some evidence that the effects of mood on learning and reconsolidation constitute distinct processes, this question remains an important issue for future work aimed at unpacking whether an WoF effect on reconsolidation is indeed mediated by arousal. (page 19, discussion)

Reviewer #2:

Summary of Main Findings

Overall, this is a well-written, well-grounded and rigorous paper. The authors found, behaviorally, that treatment condition differently modulates the effect of affect induction (WoF) on learning, such that only the SSRI group prefers items that were learned after a positive affect induction. In memory, the results suggest a more lasting memory for items encountered after the affect manipulation, and this effect was also modulated by treatment group. However, the interpretation of both results could be further developed, specifically, to consider the possibility of effects on loss aversion (see “1. substantial concerns”). The results also appear to show through computational modeling that there is a differential effect of SSRI on the m_{positive} parameter, yet it was unclear if this effect was dependent on model fitting procedures (e.g. the setting of predefined group priors). This should be clarified (see “2. substantial concerns”). Finally, although the modeling provides a potential mechanism for the delayed response of SSRI (positive feedback between affect and reward sensitivity), the case for bidirectionality could be clarified (see “3. substantial concerns”).

These results are potentially impactful, however, the substantial concerns should be addressed before proceeding.

We thank the reviewer for a positive evaluation of our study and the constructive suggestions, which we have addressed thoroughly as described below.

Substantial Concerns

1) Behavioral results in figure 3 and 4 need to be further developed. Possible alternative theories including loss aversion should be addressed.

We thank the reviewer for this valuable and intriguing suggestion. We first note that since our learning task did not entail losses, but only reward and no reward outcomes (cf. Fig. 1B), and all trials were framed in the domain of gains, loss aversion could not have directly affected task choices.

We agree that loss aversion is relevant for understanding the impact of the WoF draw, as all subjects experienced both a substantial gain and a substantial loss in the WoF draw over the course of the study. Here, loss aversion might be expected to manifest in subjects' mood ratings following the WoF draw. Indeed, we found that the absolute (negative) effect of a WoF loss on mood is significantly greater than the absolute (positive) effect of a win ($F=25.9$, $P<0.001$, cf. Fig. 5B), consistent with the idea that losses are more impactful than gains (Kahneman & Tversky, 1979). However, these mood responses were unaffected by SSRI treatment (drug: $F=0.8$, $P=0.361$; Fig. 5B), meaning that neither the mood change following a WoF win ($F=0.5$, $P=0.482$) nor the mood change following a WoF loss ($F=0.4$, $P=0.499$) differed between groups. Thus, SSRIs did not affect this manifestation of loss aversion on subjective mood ratings. Moreover, examining the downstream effects exerted by SSRIs via mood on learning, shows that it was only the impact of positive, but not negative, affect on learning that was enhanced by SSRI treatment.

We thank the referee for raising this issue, and we now address how our results, specifically the immediate effects of WoF on mood, link with loss aversion.

“A comparison of pre- and post-WoF mood (mean happiness ratings across entire block) indicated that WoF outcome profoundly impacted emotional state, i.e., subjects reported feeling happier in the post-WoF block after a win, and less happy after a loss ($F_{1,60}=42.0$, $P<0.001$). Here, the absolute (negative) effect of a loss was significantly greater than the absolute (positive) effect of a win ($F_{1,60}=23.5$, $P<0.001$, Fig. 5B), a result that has echoes with the notion of loss aversion, where a loss is known to be more impactful than an equivalent gain²³. However, these mood responses were unaffected by SSRI treatment (drug: $F_{1,60}=0.8$, $P=0.361$; Fig. 5B), meaning that neither the mood change following a WoF win ($F_{1,60}=0.5$, $P=0.482$) nor the mood change following a WoF loss ($F_{1,60}=0.4$, $P=0.499$) differed between groups. Thus, SSRIs did not affect the manifestation of loss aversion on subjective mood ratings.” (page 7, Results)

2) Priors

Were the group distributions “predefined”, as stated, or was expectation maximization used to find the group priors in a hierarchical fashion? If the former, this could bias the critical m-parameter results, so we need some evidence that the results are not sensitive to the choice of group priors, as well as clarification on where the group priors were drawn from. If the latter, were different priors fit to drug condition, or was it simply assumed that all subjects regardless of drug condition were drawn from a common distribution? If different priors were used per drug condition then any statistical test between the groups might be unduly influenced by the shrinkage imposed by the priors. Please provide more details on these methods and statistical tests used.

We apologise for a potential lack of clarity in the description of our methods.

We used an iterative expectation-maximization procedure in a hierarchical fashion. Critically, we did *not* fit different priors to different treatment groups. Instead, as inferred by the reviewer, we took a conservative approach of assuming that all subjects, regardless of treatment

condition, were drawn from a common distribution. Thus, any shrinkage imposed by the priors was identically applied to all subjects regardless of group, such that the individual subject fits are conditionally independent given the global prior which is shared across groups, and therefore does not enhance group differences. We thank the reviewer for bringing this lack of clarity to our attention. We now provide a more detailed account of the approach we used.

“Model fitting

To fit parameters of different models to subjects’ decisions, we used an iterative hierarchical expectation-maximization procedure across the entire sample^{29,55}. We sampled 10^5 random settings of the parameters from predefined prior distributions. Then, we computed the likelihood of observing subjects’ choices given each setting and used the computed likelihoods as importance weights to re-fit the parameters of the prior distributions. [...] Note that the hierarchical fitting procedure, including all priors, were applied to the entire sample without distinguishing between SSRI and placebo subjects. This ensured that the parameter estimates, at the level of individual subjects, were mutually independent given the shared prior, rendering it appropriate to assess between-group differences.” (page 33, methods)

3) Bidirectionality

Bidirectionality should be diagrammed for clarity and how the data and model supports the arrow going in each direction. I recommend that to fully argue the point of directionality, as well as how this study (time scale testing = day) can be extrapolated to the time scale of clinical effect of SSRIs (weeks), there needs to be more discussion how this positive feedback loop evolves in time. One clear prediction of the positive feedback loop would be that the as the days progress, the immediate response of the SSRI group would be significantly higher than the placebo. The claims of bidirectionality and mechanism would be weakened without this pattern.

We apologise for a lack of clarity in relation to the term “*bidirectional*” in our initial version. Previous work suggests that (positive or negative) outcomes impact upon affective state which, in turn, impacts on sensitivity to subsequent outcomes [mood \leftrightarrow reward sensitivity]. In relation to this proposed bidirectional relationship, our main result shows SSRIs specifically boost the impact of positive affect on reward sensitivity [mood \rightarrow reward sensitivity]. Thus, rather than directly impacting sensitivity to reward or mood, SSRIs influence the impact of mood on reward sensitivity, and thus enhance (or even engender; see comment #2.15 below) one direction of the putative bidirectional relationship.

Referring to the putative bidirectional relationship in the present paper is important for two reasons:

(1) if a bidirectional relationship exists then simply enhancing one of its directions, e.g., [mood \rightarrow reward sensitivity], will amplify the positive feedback loop alone. In this way, the present finding avails of previous theory (Eldar et al., 2016, *Trends Cogn Sci*) to explain why SSRIs would not only impact the effect of mood on reward responsivity (cf. revised Fig. 3B/C) but also contribute to a further downstream mood improvement.

Figure 3. SSRIs induce an impact of positive mood on learning.

[...]

(C) Interaction between affect and reward sensitivity. Rewards impact on emotional state, which, in turn, impacts on perception of subsequent rewards. In this bidirectional interaction, SSRIs enhance the impact of positive affect on reward sensitivity. This serotonergic effect can engender a positive feedback loop.

[...]

(2) the second reason for why discussing bidirectionality is important is that our result provide some evidence, albeit indirect, that can be interpreted as SSRI effects on one direction [mood \rightarrow reward sensitivity] inducing consequential downstream effects in the other direction [reward sensitivity \rightarrow mood]. Specifically, we show an enhanced delayed mood impact of a WoF win that was only evident after subjects from the SSRI group had played an additional block in which they received more rewards (cf. revised Fig. 5C/D).

Figure 5. SSRIs induce mood effects via affect-learning interaction.

[...]

(D) Interaction between affect and reward sensitivity. Rewards impact on emotional state, which, in turn, impacts on perception of subsequent rewards. Biased perception of reward, a downstream effect of a serotonergic impact on affect-learning dynamics (cf. Fig. 3B/C), can result in a delayed mood impact of the WoF that was only evident after SSRI subjects had played an additional block in which they received more rewards.

[...]

We agree with the referee that more direct evidence would require examining correlated changes in mood and reward responsivity over a longer time scale. However, the current study is not ideal for this purpose, because a sustained impact of SSRIs on mood that manifests robustly (given the random mood-influencing occurrences of daily life) can only be anticipated following a period of at least two weeks, as indicated by clinical studies. We aim to collect such longer-term data in the future (cf. response to #2.15 below).

We agree with the referee that diagrams providing clarification of the results will aid the main message of our study as well as clarify how the present findings relate to previous ideas. Thus,

we have revised the figures according to the reviewer's suggestion to show how our data support the arrows going in each direction.

Additionally, we have clarified the text concerning "*bidirectionality*", making the discussion more precise, and removing the word altogether from the abstract. We hope, this accords with the referee's suggestion.

"[...] Here, we replicate previous findings showing that unexpected outcomes impact on emotional state, which, in turn, results in subsequent outcomes being perceived as better in a positive, and worse in a negative state^{20,21}. This suggests a bidirectional relationship between affect and reward sensitivity, whereby a reward induces positive affect, and positive affect in turn renders subsequent rewards more valuable. Our novel finding is to show that serotonin boosts the impact of positive affect on reward sensitivity. SSRIs specifically enhance one axis of a two-way interaction, potentially engendering a positive feedback loop, whereby a boosted perception of positively biased rewards leads to further mood elevation over time¹⁹." (page 17/18, discussion)

Minor Concerns

4) Generalization to clinical population: is there a reason to believe that this will generalize to the clinical population. Specifically, considering this mechanism involves a positive feedback - would the clinical population have the initial increased affect to allow for this mechanism.

The reviewer raises an important point, as to the relevance for a clinical population, where an inability to experience pleasure from rewards is a core clinical symptom (Whitton et al., 2015, *Curr Opin Psychiatr*). Nevertheless, previous research shows that patients with depression do respond to experimental induction of positive mood (e.g., Falkenberg et al., 2012, *PLoS One*, Horner et al., 2014, *Depress Anxiety*; Price et al., 2017, *Biol Psychiatry*), albeit such responses are generally attenuated (cf. Bylsma et al., 2008, *Clin Psychol Rev*, for a meta-analysis) and shorter (Horner et al., 2014, *Depress Anxiety*). Critically, variability in an ability to experience positive affect predicts response to antidepressant treatment (Rottenberg et al., 2002, *Emotion*; Uher et al., 2012, *Psychol Med*), and evidence indicates emotional reactivity can vary as a function of depression subtype (Bylsma et al., 2008, *Clin Psychol Rev*; Guhn et al., 2018, *Affect Disord*).

Based on this literature, we believe that there is a reasonably high likelihood of depressed patients showing a positive affective response to a significant positive event, especially those patients likely to benefit from SSRI treatment. Precisely how a diminished response in depressed individuals relates to the downstream effects of positive affect on learning remains to be examined in future studies. Ultimately, we believe an assessment of how these effects respond to medication/therapy, and critically, how an impact on the described cognitive processes relates to clinical treatment response, will be an exciting future research avenue.

We now mention a limitation of our assessments of a non-depressed cohort, and how a reduction in the experience of positive affect in patients might impact on our findings, and we call for future studies testing our hypotheses in a clinical population.

"A limitation of our study is that we did not assess patients with clinical depression. Previous research indicates depressed individuals respond to experimental induction of

positive mood^{44,45}, albeit affective response is typically attenuated and maintained for a shorter time period^{45,46}. Critically, variability in the ability to experience positive affect predicts response to antidepressant treatment^{47,48}, and evidence indicates emotional reactivity can vary as a function of depression subtype^{46,49}. Precisely how a serotonergic impact on affective-learning dynamics relates to clinical treatment response, will need to be targeted in future clinical studies.” (page 19, discussion)

5) Abstract Sentence 3

I would suggest making a change such as adding “experimentally induced” before “positive affect on...” This would make more salient that the key effect is an interaction between induced affect change and learning/reconsolidation.

We agree with the reviewer and modified the abstract accordingly.

“Using week-long SSRI treatment in healthy volunteer participants, we show serotonin enhances the impact of experimentally induced positive affect on learning of novel, and reconsolidation of previously learned, reward associations.” (page 2, abstract)

6) Sentence 4

I’m not sure why “resulting in a delayed mood response” is in this sentence about computational modeling, as it seems this is an implication of the results as a whole, including the rule-outs of direct effects on general affect or affect responsivity to outcomes, rather than something shown directly by the modeling (although it is supported by the median split in Fig. 5D). It may also be helpful to clarify after “subjective reward perception during learning” that this was specifically following a positive (but not negative) induced affect change, as I think this would lead to a clearer segue to the next sentence.

We agree with the reviewer that the modelling per se is not showing effects of a delayed mood response (although supportive of it, as pointed out by the reviewer) and that reference to this is better placed with the following sentence. Moreover, we thank the reviewer for a second suggestion that aids the readability of the abstract’s final paragraph.

“Computational modelling indicated these effects are best accounted for by a boost in subjective reward perception during learning, following a positive, but not negative, mood induction. Thus, instead of influencing affect or reward sensitivity directly, SSRIs might amplify an interaction between the two, giving rise to a delayed mood response.” (page 2, abstract)

7) Introduction

p. 3 I found paragraph 2 somewhat hard to follow. It begins by talking about negative cognitive biases in attention and memory. Then it suggests CBT modulates such biases. I took this to mean CBT tries to decrease negative cognition (as, indeed is typically the primary focus in, eg, Beckian Cognitive Therapy). However, it then states pharmacological intervention may work similarly, in that it induces a “rosier perspective” by boosting how positive information is perceived; it’s unclear what this has

to do with negative biases. I would recommend reworking this paragraph to make the flow clearer.

One option would be to cite literature that finds decreased attention to, and interpretation in the direction of, positive information in depression (eg, Armstrong & Olantunji, 2012; Evereart, Podina, & Koster, 2017). And then to note that a target of many CBTs for depression, such as Behavioral Activation, is to increase contact with rewarding information. This I think would be a clearer segue to discussing how pharmacological treatment may similarly induce a rosier perspective.

We apologise for the missing link to behavioural therapies focusing on positive information processing. We agree with the reviewer that this link is important, particularly in light of the specific findings of our study (cf. #2.4). Indeed, our findings align well with the notion of behavioural activation in depression therapy and the aim of increasing exposure to reward (Jacobson et al., 2002, *Clin Psychol Sci*). We have included the proposed references and modified the paragraph accordingly. We hope the reviewer finds the paragraph more suitable in its revised form.

“At the cognitive level, a bias towards attending and remembering negative information⁹, as well as decrease in perception of positive information^{10,11}, are considered to play a key role in both the development and maintenance of depression. Consequently, modulating affective biases, with the aim of boosting reward exposure and positive experience, is a principled target in psychological interventions, as in the paradigmatic instance of behavioural activation therapy¹². Intriguingly, a growing body of work suggests pharmacological intervention might influence affective processing in a similar manner, in other words via a boost in perception of positive affective information that induces a “rosier” perspective^{7,13}.” (page 3, introduction)

8) p. 3 It’s unclear in what sense reward learning or memory consolidation is a “higher cognitive process” than what the authors are referring to as affective biases; consider rewording.

We apologise. In this paragraph, we refer to previous studies that assessed serotonergic effects on perceptual processing of inherently emotional information, such as face perception (Harmer et al., 2004 *Am J Psychiatry*; Norbury et al., 2009, *Psychopharmacology*; Godlewska et al., 2016, *Transl Psychiatry*). In our study, we aimed to examine whether SSRIs induce similar affective biases in cognitive process other than perception, such as, e.g., learning. We accept we did not state this precisely enough, and we have modified the paragraph accordingly.

“Nevertheless, it remains unclear how a pharmacological induction of affective biases in perceptual processing, for example as reported in emotional face perception, translates into overt mood change¹⁴⁻¹⁶. Influential theories propose affective biases may influence cognitive processes other than perception, such as reward learning and affective memory consolidation^{17,18}.” (page 3, introduction)

9) p. 9 I’m unclear on what the authors did in the logistic regression analysis. Please try to explain what’s being tested more clearly. I think it would help to put a “such that...”

after the effect of mood bias across the whole sample. Also, in Supp. Fig 1., the (B) should be placed before the text stating the effect.

&

It would be helpful to include in “Data Analysis” information about the regression analyses performed, as currently it’s unclear what statistics the authors actually ran.

In the regression model, choice on test block trials (without feedback) was the dependent variable. As predictor variables, we used the difference in number of rewards received (during preceding learning with feedback) for the two images shown on the screen, and whether these images were learned about pre- or post-WoF win or loss:

$$p(c_t = 2) = \frac{1}{1 + e^{-(\beta_0 + \beta_1 x_{1t} + \beta_2 x_{2t})}}$$

, where either the left ($c_t=1$) or right ($c_t=2$) image is chosen on trial t , x_{1t} is the difference in the number of experienced rewards during preceding learning for the right versus the left image (e.g., $9 - 3 = 6$), x_{2t} indicates whether the right versus left image was learned in a post-WoF block (coded as the WoF outcome for images learned post WoF, and 0 for images learned pre WoF).

Note that the test block only involved trials comparing images from block 1 vs. the block 2, such that the right image was *always* encountered during a different learning block than the left image.

Thus, positive coefficients for the first predictor (β_1) indicate how strongly subjects favoured images for which they had received more rewards during learning, and positive coefficients for the second predictor (β_2) indicate how strongly subjects favoured images they had encountered in a positive mood state, i.e., post-WoF after a win, and disfavoured images encountered in a negative mood state, i.e., post -WoF after a loss. β_0 is the intercept.

We thank the reviewer for prompting us to provide greater detail, as we see this as a key analysis that deserves a more detailed exposition. We have added details of these methods to the revised manuscript. Moreover, we thank the referee for the suggestion re. the legend of the supplementary figure, which we have modified accordingly.

“[...] For this, we used a logistic regression, on each session that comprised a WoF draw, that accounted not only for the impact of a mood manipulation, but also for the number of rewards subjects received for choosing each image with feedback during learning. Specifically, choice on test block trials (without feedback) was the dependent variable in the regression model on each session. Difference in number of rewards received (during preceding learning with feedback) for the two images shown on the screen, and a WoF outcome that preceded learning about each image, served as predictor variables. As expected, subjects were more likely to choose images for which they received more rewards during preceding learning (β_1 : $F_{1,60}=171.2$, $P<0.001$), but this rewards impact on choice was not affected by drug ($F_{1,60}=0.1$, $P=0.659$; Supplementary Fig. S3). In line with the average results presented above, we found in addition a significant effect of mood bias across the entire sample (β_2 : $F_{1,60}=4.0$, $P=0.049$), reflecting subjects’ preference for images they had encountered in a positive mood state after a win, and disinclination for images they had encountered in a negative mood state after a loss, an effect not driven by differences in experienced reward. Critically, this effect was modulated by drug group (drug \times valence: $F_{1,60}=4.1$, $P=0.047$), an effect driven by a significantly enhanced mood bias on the session including a WoF win ($\beta_{2_{\text{positive}}}$: $F_{1,60}=5.5$,

$P=0.022$), but not on the session including a WoF loss ($\beta_{2_{negative}}$: $F_{1,60}=0.1$, $P=0.789$) under SSRI as compared to placebo (Supplementary Fig. S3).” (page 9/10, results)

“Mood bias on learning, logistic regression [days 1 & 7]

[...] For this, we used a logistic regression, on each session that comprised a WoF draw, with choice on test block trials (without feedback) as the dependent variable on each session. As predictor variables, we used the difference in number of rewards received (during preceding learning with feedback) for the two images shown on the screen, and the block during which images were learned, i.e. pre- or post-WoF:

$$p(c_t = 2) = \frac{1}{1 + e^{-(\beta_0 + \beta_1 x_{1t} + \beta_2 x_{2t})}}$$

, where either the left ($c_t=1$) or right ($c_t=2$) image shown on the screen is chosen on trial t , x_{1t} is the difference in the number of experienced rewards during preceding learning for the left versus the right image (e.g., $9 - 3 = 6$), x_{2t} indicates whether the right versus left image was learned in a post-WoF block (coded as the WoF outcome for images learned post WoF, and 0 for images learned pre WoF). Note that the test block only involved trials comparing images from block 1 vs. the block 2. Thus, the right image was always encountered during a different learning block than the left image.

Positive coefficients for the first predictor (β_1) indicate how strongly subjects favoured images for which they had received more rewards during learning. Positive coefficients for the second predictor ($\beta_{2_{positive}}$, or $\beta_{2_{negative}}$, respectively) indicate how strongly subjects favoured images about which they had learned following a WoF win, and disfavoured images about which they had learned following a WoF loss. β_0 is the intercept.” (page 27/28, methods)

10) p. 18 Need more information about the “thorough medical screening to exclude any neurological or psychiatric interview.” Who performed screening and what tools (eg, structured interview) were used? When was the screening given?

Subjects were screened over the phone, by a member of the team, and assessed by a medical doctor at the study location prior to the first session. Assessment included an electrocardiogram, a medical history, including previous and current illness, prior allergic drug reactions and chronic medication intake. This assessment followed a specified written protocol. We did not use a structured psychiatric interview as this was redundant due to the fact that no participant reported concurrent psychological symptoms indicative of disorder or a history of neuropsychiatric illness. However, subjects were asked to fill out multiple pen-and-paper questionnaires assessing current psychiatric symptoms, as specified in the manuscript. We have modified the revised manuscript that now includes more detailed information about the assessment before study enrolment.

“Before study enrolment, subjects were medically screened over the phone, and assessed by a medical doctor at the study location upon arrival for the first study session. This assessment included an electrocardiogram to exclude QT interval prolongation, as well as a structured medical history, to ensure subjects had no current health issue, no history of neurological or psychiatric disorder, no prior allergic reaction to any drug, and were not taking any regular medication.” (page 20, methods)

11) It would be helpful to explain briefly what subjects were told re: the possibility of getting SSRI or placebo.

We added more clarification to the revised manuscript.

“Subjects were informed that there was a 50% chance that they would be in the active drug group, and a 50% that they would be allocated to the placebo group. Additionally, subjects were informed regarding the double-blind procedure of the study, meaning that they were not told which group they had been allocated to, and that the study team responsible for data collection was also unaware about their drug group allocation.” (page 21, methods)

12) p. 20 The sentence beginning “Note that the session IV test block...” seems like it should go in the preceding section, “Learning blocks.”

In fact, the sentence referred to by the reviewer was deliberately placed in this section. This is because the test block on session IV, in contrast to the previous sessions, comprised 24 additional trials. This design feature allowed us to assess the impact of the mood manipulation on reconsolidation of reward associations.

Specifically, on these trials, subjects were asked to choose between images of similar reward probability, i.e., high vs. high, and low vs. low, that were learned on session III, but encountered again as interleaved maintenance trials during the learning blocks on sessions IV. Thus, this set of trials served to assess whether the mood manipulation induced a specific preference for images learned during the same block, but encountered again during different blocks, before vs. after the WoF draw.

We understand that the design is complex. Importantly, design features were specifically implemented in this way to allow testing of our key hypotheses. Thus, to improve understanding of the design’s rationale, we have added an additional figure to the supplementary material that depicts the structure and the order of image appearance within each session in more detail. We hope this can aid understanding as well as facilitate replication of our design.

Moreover, we have added more information about this additional set of test block trials, which distinguishes the session IV test block from test blocks on previous sessions, to the revised methods section.

“Note that the session IV test block comprised 72 trials, containing 24 additional trials, on which subjects were asked to choose between images of similar reward probability, i.e., high vs. high, and low vs. low. Critically, this set of trials comprised images that were learned in the same block on the previous session, i.e., session III, but encountered as interleaved maintenance trials either before or after the WoF draw on the current session, i.e., session IV. This design feature allowed us to assess the impact of mood on reconsolidation of reward associations, and is described in detail below.” (page 23)

session	block I	block 2	test	learning
I.a	1-4	5-8		maintenance test
I.b	9-12 1-2 & 5-6	13-16 3-4 & 7-8	9-16	
II	17-20 9-10 & 13-14	WoF 11-12 & 15-16	21-24	17-24
III	25-28 17-18 & 21-22	29-32 19-20 & 23-24		25-32
IV	33-36 25-26 & 29-30	WoF 27-28 & 31-32	37-40	33-40 25-32

Supplementary Figure 1. *Experimental design: session structure and image appearance.*

All subjects were exposed to an identical set of 40 images in the study, where image order (1-40) was randomly shuffled across participants.

Each session comprised two learning blocks (image numbers depicted in orange), where subjects encountered four novel images, two with a high (70%) and two with a low reward probability (30%). Feedback enabled subjects to learn about images' reward probabilities through trial-and-error. After the second learning block, subjects performed a test choice block (depicted in blue), where they chose between images from the two preceding learning blocks. Maintenance performance of learned reward associations was assessed during interleaved trials (depicted in olive) involving familiar images on subsequent sessions, where images learned about in session I re-appeared in session II, and images from session II in session III, etc. Notably, both learning and maintenance were assessed on choices between the same image pairs, with the time-lag between learning and maintenance test constituting the only difference. To prevent further learning, outcomes were not revealed in test block *and* maintenance trials, and subjects were instructed to rely solely on what they had learned previously about the images.

Additionally, sessions II and IV comprised a WoF draw between learning blocks, where each subject either won or lost a large monetary amount.

Session IV test block comprised 24 additional trials, on which subjects were asked to choose between images that were learned in the same block on the previous session, i.e., session III, but encountered as interleaved maintenance trials either before or after the WoF draw on the current session, i.e., session IV. This design feature allowed us to assess the impact of mood on reconsolidation of reward associations.

Session I included two additional learning blocks to familiarise subjects with the task. Thus, the session comprised two learning blocks during the first part (I.a), and two learning blocks during the second part (I.b), i.e., four in total.

Mood bias on learning was assessed on session II (images 17-20 vs. 21-24), and IV (33-36 vs. 37-40). Mood bias on reconsolidation was assessed on session IV (25-26 vs. 27-28, and 29-30 vs. 31-32).

13) p. 23 Sentence beginning “Since all images re-appeared..” I’m having difficulty following what this is referring to—is it the interleaved maintenance trials? It would be helpful to clarify.

The reviewer is correct that the sentence refers to the interleaving of maintenance trials during learning blocks. This design feature allowed us to test session-to-session maintenance of learned reward associations. We hope that the additional supplementary figure, depicting the

sessions' structure (cf. #2.13), facilitates clearer understanding of this issue. Additionally, we have specified this sentence according to the reviewer's suggestion.

“Since all images reappeared on interleaved maintenance trials, without feedback, during the subsequent session, subjects needed to retain learned reward associations to perform well.” (page 26, methods)

14) Results

Figure 3 title: “SSRIs boost the impact of positive mood on learning” - this implies that there is some baseline impact of positive mood on learning - however based on the fact that there is no preference in the placebo group when there is a win, it seems that this baseline does not exist.

We agree with the reviewer that the wording is misleading. We have modified the figure title accordingly.

“SSRIs induce an impact of positive mood on learning.” (figure 3)

15) Discussion

This theory seems to make precise experimental predictions. If the authors have space, I'd recommend noting some of these as a guide for future research about how their claims could be tested / falsified. For example, the theory seems to predict that as depressed individuals start an SSRI, event-contingent increases in good mood should lengthen gradually over time, mediated by intervening positive learning experiences. Moreover, how much good moods are lengthened should be moderated by individual differences in the M_positive parameter. The latter could be tested by combining computational modeling and EMA, and the former could be tested just using EMA.

We are pleased about this comment. Indeed, these are exactly the kind of research questions we aim to pursue in future research. We briefly mentioned this in the initial version, but now elaborate on this aspiration with a specific paragraph, depicting outstanding research questions that arise from this study that can be tested/falsified. Please see also changes mentioned in #2.4 with regard to predictions re. an impact on positive or negative biases in depression.

“[...] Moreover, we acknowledge that week-long SSRI treatment does not typically exert a clinically meaningful impact on depressive symptoms²². Longitudinal study designs, over longer timescales, that probe whether interindividual variability in candidate computational parameters (e.g., the influence of positive affect on learning) relates to clinical treatment response are an avenue for future research. Thus, studies combining pharmacological intervention, repeated behavioural testing, and ecological momentary mood assessment, might reveal whether an early serotonergic impact on affect-learning dynamics (within hours or days) provides a useful predictive marker for clinical response (after multiple weeks)^{14,50,51}.” (page 20, discussion)

Reviewer #3:

The authors demonstrated week-long SSRI treatment enhanced the impact of positive affect on learning of novel, and reconsolidation of previously learned, reward associations, by a placebo-controlled study. Based on the result, they concluded SSRIs amplify a bilateral interaction between affect and reward sensitivity instead of influencing them directly. The main research findings of this paper will be quite important for understanding of serotonin's delayed antidepressant effects. However, I have to express several concerns as below.

Major points

1) I think the authors combined the 70%vs70% and 30% vs30% tests. Didn't author analyze 70%vs70% and 30% vs30% tests separately? In my opinion, 70% and 30% images themselves might contain positive and negative valence, respectively.

The reviewer raises an interesting issue here. She/he is right that we collapsed high vs. high [70% vs. 70%], and low vs. low [30% vs. 30%] probability trials, when analysing trials with images of equal reward probability. Here, we are following a wide convention in this type of learning study which typically use distinct probabilities. We are not aware of any a priori body of work that suggests these different probabilities are associated with different levels of positive or negative affect. The underlying rationale of this analysis, that we used in our previous study (Eldar et al., 2015, *Nat Commun*), is that we assessed choices between images of equal reward probability, irrespective of the reward probability per se. Another key reason we did not compute separate analyses is substantial reduction of statistical power when dividing the data into two sub-comparisons.

However, we have followed the reviewer's suggestion and compared the results of the two separate sub-comparisons, i.e., [70% vs. 70%] vs. [30% vs. 30%]. Here, we found no significant difference when comparing the high [70% vs. 70%] versus the low probability [30% vs. 30%] results, for either the win or loss condition, across the whole sample (win: $F=0.9$; $P=0.350$, loss: $F=3.2$, $P=0.079$), and also no effect in either treatment group separately ([SSRI] win: $F=0.5$, $P=0.478$, loss: $F=2.4$, $P=0.132$; [placebo] win: $F=0.4$, $P=0.540$, loss: $F=0.09$, $P=0.356$).

An equivalent analysis for the reconsolidation results similarly revealed the absence of a difference when comparing the high [70% vs. 70%] versus the low probability [30% vs. 30%] results (win: $F=0.06$, $P=0.801$, loss: $F=0.3$, $P=0.583$), [SSRI] win: $F=0.02$, $P=0.885$, loss: $F=0.04$, $P=0.844$; [placebo] win: $F=0.04$, $P=0.835$, loss: $F=0.3$, $P=0.592$).

We now present these sub-analyses in the methods section of the revised manuscript.

“Crucially, this analysis was conducted for test block trials that included image pairs with objectively similar reward probabilities that had been encountered in different blocks. There was no significant difference for [high vs. high] vs. [low vs. low] for the win ($F_{1,60}=0.9$, $P=0.350$) or loss ($F_{1,60}=3.2$, $P=0.086$) conditions. Thus, we collapsed trials, i.e., high vs. high, and low vs. low, to increase statistical power for the analysis of trials comprising images pairs with similar reward probabilities.” (page 26, Methods)

“This analysis comprised test block trials comparing image pairs with objectively similar reward probabilities that had been re-encountered in different blocks. There was no

significant difference for [high vs. high] vs. [low vs. low] for the win ($F_{1,62}=0.06$, $P=0.801$) or loss ($F_{1,62}=0.3$, $P=0.583$) conditions. Thus, we collapsed trials, i.e., high vs. high, and low vs. low, to increase statistical power for the analysis of trials comprising images pairs with similar reward probabilities.” (page 29, Methods)

2) In figure 4B (page 12), subjects preferred images encountered after losing but not winning the draw in the placebo group. The fact might not correspond to the fact described in figure 3B; More detailed explanations from the perspective of reward associations would be needed. Why did participants prefer post-WoF after loss? They tried to interpret from the perspective of “arousal”, which is quite ambiguous.

We apologise for a lack of clarity in the description of the results. A similar issue was raised by reviewer #1.

We have conducted additional analyses to probe whether the WoF acts to increase the value of images re-encountered during post-WoF blocks, or subjects simply remember those post-WoF images more accurately. Our design allowed us to rule out the latter interpretation. When we assess whether performance (on high vs. low reward probability trials) differed between images re-encountered during the post-WoF and the pre-WoF block, we find no significant difference (post-WoF vs. pre-WoF: $F=1.2$, $P=0.302$). Moreover, we found no evidence that subjects performed more accurately as a function of valence of the draw (post-WoF win vs. post-WoF loss: $F=0.4$, $P=0.538$), or as a function of drug condition (post-WoF drug: $F=0.04$, $P=0.845$, post-WoF drug x valence: $F=0.7$, $P=0.417$). These results rule out an interpretation based upon images being remembered more accurately either as a function of mood induction or of drug condition.

The finding for reconsolidation in placebo subjects is puzzling at first sight. However, interpretation here is aided by the fact that across the entire sample, there is a fundamental difference between the effect of WoF on learning and reconsolidation (cf. new supplementary figure below). Specifically, overall, subjects preferred images reconsolidated during the post-WoF block. Thus, in contrast to the effect on learning, this latter effect is consistent with a valence-independent boost in subjective value of post-WoF images, possibly mediated through the enhancement of arousal by the WoF draw (since, overall, this effect was evident following both positive and negative mood induction). This difference between the two effects of the WoF, on learning and on reconsolidation, is further substantiated by the absence of any relationship between the two (win: $R=0.07$, $P=0.705$; loss: $R=-0.18$, $P=0.303$), suggesting the two constitute distinct processes.

It is an open question why positive and negative mood induction would both serve to increase the subjective values of re-appearing images. One favoured interpretation is that this effect was mediated by increased arousal. Importantly, SSRIs modulated the effect of the WoF on reconsolidation in the same way as they modulated the WoF effect on learning. That is, in both cases SSRIs boosted the impact of positive mood on subjective value, as compared to the impact of negative mood. Thus, overall, the results indicate that winning the WoF draw enhances subjective value of images encountered after the draw, and this effect is enhanced by a SSRI. Critically, these results align with our a priori predictions, namely, that SSRIs act to strengthen (or even induce) the downstream impacts of positive affect.

We apologise for not providing enough information on this secondary finding. We have now added the additional analyses and clarifications to the revised manuscript. Additionally, we now present additional figures in the supplementary material that will hopefully aid understanding of the results.

“Across the entire sample, subjects preferred images that re-appeared post-WoF, irrespective of whether they were linked to a win or loss (assessed on equal reward probability trials; $F_{1,60}=7.5$, $P=0.008$; Fig. 4B). This raises a possibility that increased arousal, engendered by the outcomes of the WoF draw, enhanced the subjective value of images during post-WoF reconsolidation, inducing a consequential preference for these images. This point to a possibility of a qualitatively distinct effect of the WoF draw on learning and on reconsolidation (Supplementary Fig. S2 for effects across the entire sample), a difference that is further supported by the absence of any relationship between the two effects (win: $R=0.07$, $p=0.705$; loss: $R=-0.18$, $P=0.303$).

Supplementary Figure 2. Effect of mood on learning and reconsolidation, across entire sample.

(A) Proportion of test block choices between images encountered during preceding *learning* in different blocks, i.e., pre-WoF or post-WoF. This analysis compared choices between images with equal reward probabilities, i.e., high vs. high and low vs. low, and preference is computed as difference between percent choices of post-WoF images and pre-WoF images, as a function of WoF outcome.

Participants showed a preference for images encountered in better mood, favouring post-WoF images after a winning draw, and favouring pre-WoF images after a losing draw.

(B) Proportion of test block choices between images that had been encountered without feedback during preceding *reconsolidation* in different blocks, i.e., pre-WoF or post-WoF. This analysis involved choices between images with objectively similar reward probabilities, i.e., high vs. high and low vs. low, and preference is computed as difference between percent choices of post-WoF images and pre-WoF images, as a function of WoF outcome.

Contrasting a mood bias on learning, subjects preferred images that re-appeared post-WoF, irrespective of whether the WoF was won or lost. ** $p<0.01$, * $p<0.05$, (*) $p=0.1$. Error bars indicate SEM.

Critically, an arousal bias was differentially modulated by treatment (drug \times valence: $F_{1,60}=4.8$, $P=0.032$; Fig. 4B), where the SSRI group showed a significant preference for images encountered after winning, but not after losing, (win: $F_{1,15}=6.4$, $P=0.023$; loss: $F_{1,15}=1.4$, $P=0.246$); and the placebo group after losing but not winning (win: $F_{1,15}=0.5$, $P=0.819$; loss: $F_{1,15}=10.1$, $P=0.006$).

We found no evidence that subjects performed more accurately (assessed on high vs. low probability trials), as a function of the draw (post-WoF vs. pre-WoF: $F_{1,62}=1.3$, $P=0.301$), valence of the draw (post-WoF win vs. post-WoF loss: $F_{1,60}=0.4$, $P=0.538$), or drug condition (post-WoF drug: $F_{1,60}=0.04$, $P=0.845$, post-WoF drug \times valence: $F_{1,60}=0.7$, $P=0.417$). These observations rule out an interpretation based upon images being remembered more accurately as a function of mood induction or of drug condition.

Thus, the findings are consistent with the WoF draw amplifying the subjective value of images re-encountered in the post-WoF block, leading to a biased choice preference towards

such images. As predicted, SSRI treatment induced a preference for images re-encountered in a positive, compared to a negative, mood state.” (page 13/14, results; and supplementary material)

We have added further discussion of this result to the revised manuscript:

“Another limitation pertains to the mechanistic understanding of a secondary finding, where positive and negative affect boosted subjective value during post-WoF reconsolidation. Although our study provides some evidence that the effects of mood on learning and reconsolidation constitute distinct processes, this question remains an important issue for future work aimed at unpacking whether an WoF effect on reconsolidation is indeed mediated by arousal.” (page 19, discussion)

Minor points

3) In my opinion the description of the methodology and technical details should be shortened considerably. The experimental procedures of the scheme could be summarized in a table instead of being described in the text.

We followed the reviewer’s suggestion and shortened description of technical details in the results section of the main manuscript. Instead, we now refer to a more detailed description in the methods section, so as to enable the reader to follow the experimental procedures in detail and replicate them. Moreover, we have added a detailed description of the experimental design, containing the order of image appearance across sessions, in a separate figure in the supplementary material (cf. #2.13 & #2.14).

4) The introduction and discussion sections seem to include several repetitions which could be shortened.

It is difficult to exactly pin down to what passages the reviewer is referring to here, and whether she/he is referring to a repetition within introduction, or between introduction and discussion. We have shortened two specific paragraphs based on this suggestion, and hope the reviewer agrees with these modifications.

5) In discussion part, it seems inadequate to clearly conclude that these findings fill a previously missing link (page 16, line9-11), only from written results. Seven days of research might not be enough to generalize the theory as the authors mentioned in the limitation part. What "an early marker" in the last sentence of the discussion part refers to is unclear.

We apologise. In this paragraph, we refer to previous studies assessing the impact of antidepressants on perceptual processing of inherently emotional information, such as, e.g., visual face perception (e.g., Harmer et al., 2004 *Am J Psychiatry*; Norbury et al., 2009, *Psychopharmacology*; Godlewska et al., 2016, *Transl Psychiatry*).

In our study, we show that SSRIs similarly induce affective biases in cognitive processes other than perception, such as learning and reconsolidation. Critically, mood disorders are often characterised by aberrant sensitivity to, and an inability to accurately learn from, reward (Vrieze et al., 2013, *Biol Psychiatry*; Whitton et al., 2015, *Curr Opin Psychiatry*).

We show SSRI-induced affective biases extend to cognitive processes whose dysfunction is thought to underpin the emergence and maintenance of depression. In this way, our study provides a step towards understanding how antidepressants impact upon dysfunctional cognition in depression. We modified the respective paragraphs and removed the potentially misleading phrase of a "missing link".

"[...] First, previous research suggests serotonergic antidepressants induce a bias in the perception of inherently emotional information, for example emotional face perception^{15,16}. We show that SSRI-induced affective biases influence cognitive processes other than perception, such as learning and reconsolidation. Critically, mood disorders are often characterised by aberrant sensitivity to, and an inability to accurately learn from, reward^{32,33}. Our study is a step towards understanding how antidepressants impact upon dysfunctional cognition in depression, and how a relatively immediate serotonergic effect on implicit affect-cognition interactions can set the scene for a mood improvement that emerges over time¹⁷. (page 18, discussion)

In relation to how the effects might provide an early marker of treatment response, our results indicate that changes in affect-learning dynamics occur within the first days of an intervention. Based on previous findings (e.g., Tranter et al., 2009, *J Affect Disord*; Shiroma et al., 2014, *Int J Geriatr Psychiatry*, Godlewska et al., 2016, *Translat Psychiatry*), we hypothesize such early serotonergic effects on cognition (seen within the first week of treatment, i.e., preceding explicit changes in mood) might predict a clinical mood response in later stages of treatment (on a 4-8 week time scale). Thus, they could serve as an early predictive marker of treatment responsiveness by discriminating those who might, or might not, profit from treatment.

We agree with the reviewer that further research, using longer pharmacological protocols, and, importantly, assessing patients with mood disorders, will be needed to establish a causal link between a serotonergic impact on the relationship between mood and reward sensitivity and treatment response. Thus, we have now modified the paragraph and added a more detailed limitations section, in combination with a call for specific future studies, to the revised manuscript (please also see comments to #2.4 & #2.16 and respective changes in the revised manuscript).

"[...] Moreover, we acknowledge that week-long SSRI treatment does not typically exert a clinically meaningful impact on depressive symptoms²². Longitudinal study designs, over longer timescales, that probe whether interindividual variability in candidate computational parameters (e.g., the influence of positive affect on learning) relates to clinical treatment response are an avenue for future research. Thus, studies combining pharmacological intervention, repeated behavioural testing, and ecological momentary mood assessment, might reveal whether an early serotonergic impact on affect-learning dynamics (within hours or days) provides a useful predictive marker for clinical response (after multiple weeks)^{14,50,51}." (page 20, discussion)

"[...] and hold great promise as an early marker of treatment response that can aid tailoring of currently available, and development of novel, therapeutic approaches to depression." (page 20, discussion)

Reviewers' Comments:

Reviewer #1:

Remarks to the Author:

The authors have addressed all my points carefully and I am happy to sign off on this.

Reviewer #2:

Remarks to the Author:

Summary: The authors did a good job going through comments and addressing points made by all reviewers. While most of the comments were addressed, there are still some points that need to be clarified.

Overall, the authors did a good job acknowledging the limitations and notwithstanding these, this paper has potential to be high impact.

Major Points:

For Figure 5 and the associated statistics in the text, it's unclear why the 3-way interaction of time (immediate vs. delayed) * drug * WoF valence is not reported. It appears that SSRI leads to sustained effect of + WoF on mood over time, but also enhanced regression over time toward baseline mood (blunted reaction to the negative WoF or faster recovery from negative experience). Regardless of the significance, 5B and 5C should be connected and discussed together (if underpowered to detect a 3-way interaction, can acknowledge this as a limitation).

The authors instead report an interaction within delayed mood. There they report a drug*WoF valence interaction, but not specific contrasts (significant amplified positive mood following WoF win or blunted decrease in mood, ie, decreased loss aversion, following WoF loss). Figure 5C suggests both effects are driving the interaction. Without showing the interaction effect is driven specifically by amplified positive mood, it is unclear why the discussion/takeaways of the paper focuses specifically on positive mood amplification.

As pointed out by the authors, the clinical effect of SSRIs is not evident until many weeks later. The authors discuss how the positive loop in this mechanism may relate to this delayed effect ('SSRIs specifically enhance one axis of a two-way interaction, potentially engendering a positive feedback loop'). The point may be strengthened by including words such as nonlinear and explaining how this relates with the proposal that the time course of SSRIs is delayed due to proposed bidirectional mechanism.

Minor Points:

Supplementary Figure 1 is helpful to clarify the study structure. It would probably help to add the absolute day of the study in the session column; otherwise it's easy for the reader to confuse the 3 time scales of the study (block, session, day).

Figure 5E is mislabeled as D in the figure text.

Reviewer #3:

Remarks to the Author:

I have read with great interest the response letter and the revised version of authors' manuscript. The manuscript has been adequately revised in response to my comments. Authors' explanations and revisions are satisfactory.

We thank the referees for the constructive review process, and their positive evaluation of our revised manuscript “A mechanistic account of serotonin’s impact on mood” (NCOMMS-19-1125295A). We addressed the remaining concerns below. The respective changes in the manuscript are marked in bold. Please note that the page numbers given below refer to the revised manuscript.

Reviewer #1:

The authors have addressed all my points carefully and I am happy to sign off on this.

We thank the reviewer for the appreciation of our work and their previous suggestions that helped us to substantially improve the manuscript.

Reviewer #2

Summary: The authors did a good job going through comments and addressing points made by all reviewers. While most of the comments were addressed, there are still some points that need to be clarified.

Overall, the authors did a good job acknowledging the limitations and notwithstanding these, this paper has potential to be high impact.

We thank the reviewer for the positive evaluation of our work and the kind words re. its putative impact. We have addressed the remaining concerns as described below.

1) For Figure 5 and the associated statistics in the text, it’s unclear why the 3-way interaction of time (immediate vs. delayed) * drug * WoF valence is not reported. It appears that SSRI leads to sustained effect of + WoF on mood over time, but also enhanced regression over time toward baseline mood (blunted reaction to the negative WoF or faster recovery from negative experience). Regardless of the significance, 5B and 5C should be connected and discussed together (if underpowered to detect a 3-way interaction, can acknowledge this as a limitation).

The reviewer raises an interesting issue. We followed the suggestion to compute an ANOVA with three levels, adding the factor ‘time’ (levels: 1st/2nd/3rd block), to the factors ‘drug’ and ‘valence’, respectively. This analysis does not reveal a significant 3-way interaction ($P=0.172$). Critically, this result does not conflict with an effect of SSRIs on mood per se, but merely indicates the sustained drug effect was not significantly greater than the immediate effect. However, as the referee suggests, this may be due to a lack of power to detect such a subtle effect. We now acknowledge this limitation along with a revised, more cautious interpretation of this finding (see also response to #2.2).

“Note that we did not find a significant three-way interaction (factors ‘time’: 1st/2nd/3rd block, ‘drug’, ‘valence’; $F_{2,122}=1.8$, $P=0.172$). This does not conflict with an effect of SSRIs on mood per se, but merely indicates the delayed drug effect was not significantly greater than the immediate effect. However, the finding of a delayed effect needs to be interpreted with caution.” (page 16, results)

2) The authors instead report an interaction within delayed mood. There they report a drug*WoF valence interaction, but not specific contrasts (significant amplified positive mood following WoF win or blunted decrease in mood, ie, decreased loss aversion, following WoF loss). Figure 5C suggests both effects are driving the interaction. Without showing the interaction effect is driven specifically by amplified positive mood, it is unclear why the discussion/takeaways of the paper focuses specifically on positive mood amplification.

The reviewer is right that both effects are driving the delayed SSRI mood effect, i.e., a decreased effect for losses, and an amplified effect after wins, in SSRI, compared to placebo, subjects. We have added this information to the revised manuscript.

“Indeed, we found a distal WoF-impact was differentially modulated by drug. ($F_{1,60}=5.6$, $P=0.020$). Note that this effect was not attributable to a significant drug effect for the win ($F_{1,60}=1.7$, $P=0.189$), or for the loss condition ($F_{1,60}=0.28$, $P=0.095$), but driven by an interaction of an amplified positive, and lower negative mood response in the SSRI as compared to the placebo group (Fig. 5C).” (page 15)

The main reason the discussion is specifically focused on an effect of positive mood is that this effect fits conceptually and statistically with a similar effect of positive mood that we find with regards to choice behaviour. In particular, as shown in figure 5E, there is a strong relationship between a positive mood bias and a positive delayed mood response. However, we agree this finding should be discussed more cautiously, and thus, we have modified the description and toned down the interpretation of this finding in the revised manuscript.

“Overall, SSRIs did not influence affect directly, but caused a delayed mood response, putatively a downstream effect of altered responsivity to reward. Note, however, a downstream effect, as a function of biased learning, is only one possible interpretation that requires further investigation.” (page 17)

“In conclusion, we show that SSRIs boost positive affective biases in reward processing, an effect that may result in a delayed mood response.” (page 20)

3) As pointed out by the authors, the clinical effect of SSRIs is not evident until many weeks later. The authors discuss how the positive loop in this mechanism may relate to this delayed effect (‘SSRIs specifically enhance one axis of a two-way interaction, potentially engendering a positive feedback loop’). The point may be strengthened by including words such as nonlinear and explaining how this relates with the proposal that the time course of SSRIs is delayed due to proposed bidirectional mechanism.

We thank the reviewer for this valuable suggestion that aligns with our interpretation of the results. We have modified the respective paragraph accordingly.

“SSRIs specifically enhance one axis of a two-way interaction, potentially engendering a positive feedback loop, whereby a boosted perception of positively biased rewards leads to further mood elevation¹⁹. Such non-linear dynamics may result in the gradual co-evolution of positive mood and a preponderance of positive experiences, thereby giving rise to the delayed clinical onset of SSRI therapy²².” (page 18)

4) Supplementary Figure 1 is helpful to clarify the study structure. It would probably help to add the absolute day of the study in the session column; otherwise it’s easy for the reader to confuse the 3 time scales of the study (block, session, day).

The reviewer raises a valuable suggestion. We have added an additional column, indicating study day, adding more precision to the figure re. the study’s different time scales.

day	session	block I	block 2	test	learning maintenance test
I	I.a	I-4	5-8		
I	I.b	9-12 I-2 & 5-6	13-16 3-4 & 7-8	9-16	
I	II	17-20 9-10 & 13-14	WoF 11-12 & 15-16	21-24	17-24
4	III	25-28 17-18 & 21-22	29-32 19-20 & 23-24	25-32	
7	IV	33-36 25-26 & 29-30	WoF 27-28 & 31-32	37-40	33-40 25-32

5) Figure 5E is mislabeled as D in the figure text.

We thank the reviewer for bringing this mistake to our attention. We corrected it accordingly.

Reviewer #3:

I have read with great interest the response letter and the revised version of authors' manuscript. The manuscript has been adequately revised in response to my comments. Authors' explanations and revisions are satisfactory.

We are grateful to the reviewer for the constructive review process that led to an improved version of our manuscript.